# Review of Advanced Medical Telerobots

**Sarmad Mehrdad** [1,†], **Fei Liu** [2,†], **Minh Tu Pham** [3], **Arnaud Lelevé** [3,*] **and S. Farokh Atashzar** [1,4,5]

1   Department of Electrical and Computer Engineering, New York University (NYU), Brooklyn, NY 11201, USA; sm9167@nyu.edu (S.M.); sfa7@nyu.edu (S.F.A.)
2   Advanced Robotics and Controls Lab, University of San Diego, San Diego, CA 92110, USA; f4liu@ucsd.edu
3   Ampère, INSA Lyon, CNRS (UMR5005), F69621 Villeurbanne, France; minh-tu.pham@insa-lyon.fr
4   Department of Mechanical and Aerospace Engineering, New York University (NYU), Brooklyn, NY 11201, USA
5   NYU WIRELESS, Brooklyn, NY 11201, USA
*   Correspondence: arnaud.leleve@insa-lyon.fr; Tel.: +33-0472-436035
†   Mehrdad and Liu contributed equally to this work and share the first authorship.

**Abstract:** The advent of telerobotic systems has revolutionized various aspects of the industry and human life. This technology is designed to augment human sensorimotor capabilities to extend them beyond natural competence. Classic examples are space and underwater applications when distance and access are the two major physical barriers to be combated with this technology. In modern examples, telerobotic systems have been used in several clinical applications, including teleoperated surgery and telerehabilitation. In this regard, there has been a significant amount of research and development due to the major benefits in terms of medical outcomes. Recently telerobotic systems are combined with advanced artificial intelligence modules to better share the agency with the operator and open new doors of medical automation. In this review paper, we have provided a comprehensive analysis of the literature considering various topologies of telerobotic systems in the medical domain while shedding light on different levels of autonomy for this technology, starting from direct control, going up to command-tracking autonomous telerobots. Existing challenges, including instrumentation, transparency, autonomy, stochastic communication delays, and stability, in addition to the current direction of research related to benefit in telemedicine and medical automation, and future vision of this technology, are discussed in this review paper.

**Keywords:** teleoperation; medical robotics; share autonomy; multilateral telerobotics; telerehabilitation; telesurgery

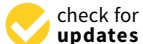



## 1. Introduction

### 1.1. Telerobotics General Context

Telerobotic systems have extended the sensorimotor capacity of humans beyond the natural competence to achieve an augmented sensorimotor ability, which allows humans to interact with objects and environments remotely [1]. During the last two decades, teleoperated robotic systems have attracted a great deal of interest due to the remarkable benefit of medical [2,3], and non-medical applications [4]. Focusing on medical applications, two major categories can be identified, namely, teleoperated surgery [2,5,6], and teleoperated rehabilitation robotic systems [7,8], which have shown significant potential to transform the delivery of healthcare services. In this regard, teleoperated surgery has been widely investigated and commercialized. Yet, telemedicine applications also include various new remote services (whose interest grows during pandemics), such as a simple consultation or an expertise with a specific healthcare professional, telemonitoring/diagonosis [9], or an assistance for a particular procedure or examination [10–19], over telecommunication networks.

Conventional telerobotic systems are composed of one leader robotic module (classically named as a master module) and one follower robotic module (classically named

as a slave module) [1]. The leader robot is to be operated by the human operator, and the follower robot is to replicate the motions of the leader robot for interaction with the environment. The potential advantages of robotic manipulation over human manipulation are numerous and one can mention:

- agility; precision; repeatability;
- automatic trajectory tracking and no-fly-zone generation;
- ability to satisfy constraints in position, and speed domains;
- real-time fusion of multimodal exteroceptive information;
- automatic recording of gestures made.

### 1.2. Introduction of Robotics in Medicine

The rise of robotics in medicine has opened up broad perspectives and suggests further progresses, not only related to the precision and comfort of surgeons but also related to the benefits for patients (reduction of invasiveness, recovery time, reduction in pain, and side effects). The medical robot remains a cooperative device for the practitioner, the sole leader on board. Moreover, humans cannot be replaced by robots in the context of medicine (in particular surgery) due to several factors, some of which relate to the need for extremely high cognitive-sensorimotor skills of the clinicians (e.g., surgeons) and imperative medical domain knowledge which cannot be replaced (this is a controversial topic). Medical doctors should be at the heart of the operation because they can integrate complex information from multiple sources and conduct the operation based on their complex medical knowledge related to the physiological and pathological context of the operation. They also possess not only the ability to analyze a rare situation and make critical decisions but also the ability to adapt, even improvise in rare cases. For all these reasons, it matters to keep the practitioners as much as possible in the loop while trying to augment their cognitive and sensorimotor competence using multimodal telerobotic systems.

In the context of surgery, the environment is usually the organs of the patient on which the surgery is being operated. This is the feedforward information path. As the feedback path, the operator will receive sensory information from the environment about the interaction between the follower robot and the environment, through various sensory channels, e.g., haptics, visual, auditory. The sensory information replicated by the leader robot for the operator allows her/him to conduct the operation based on transparent situational awareness achieved using the received sensory information [2,20–23]. Medical applications of various extensions of conventional topology have also been explored in the literature and will be discussed in this paper in Section 2.

### 1.3. History of Telerobotics in Medicine

The first medical robot was developed in Vancouver B.C., Canada, in 1983, while the first use of a (non-medical) robot for medical purpose happened on 11 April 1985, at the Memorial Medical Center of Long Beach (California) during a stereotactic brain surgery [24]. During this kind of operation, a stereotactic probe is manually inserted in the brain so that its tip is exactly positioned at the location of a tumor located employing a computerized tomographic (CT) scanner. The surgeon can use this probe as a frame to safely perform a biopsy or any other neurosurgery operations. Yet, the insertion of the probe through a straight-lined trajectory required much experience to limit the lateral damages. In this application, a Puma 200 6-degree-of-freedom (DoF) industrial robot (Unimation) held this probe and performed a linear trajectory to install the probe according to the target position and the approach orientation determined from the CT pictures. See [25] for an overview of the use of industrial robots in surgery.

The first medical robots (specifically manufactured for medical application and not a modified industrial robot) have been introduced in operating rooms a few years later (in the late 1980s and early 1990s), mainly to assist surgeons for needle placement with the patient located inside an imaging system (see [26] for a recent detailed historical review),

but their rise happened with teleoperated robots for Laparoscopy, where the need of controlling the endoscope was the first widespread robotic application. To understand this craze, it is necessary to distinguish between remote operation robots, guidance robots, telerehabilitation, and robotic simulators for training hospital staff [27,28].

In 2001, Sung et al. stated that laparoscopy had changed the way surgeons perceive and practice surgery [29]. Indeed, laparoscopy was a new surgery technique, under the umbrella of Minimally Invasive Surgery (MIS), which involved only small incisions in the abdomen to install a laparoscope surgical tool from the outside of the patient's body. The main benefits of this technique, such as reduced blood loss [30], reduced damage to the tissues [31], reduced risk of infection [32], reduced recovery time [33], and cosmetic benefits [34], were provided for the patients while it was also a new challenge for surgeons to manipulate such tools. The limited sensory awareness has been a conventional bottleneck for manual MIS [35,36]. It led to the need for extensive and specialized sensorimotor training; however, even with such training, experienced surgeons had difficulties in efficiently perceiving the surgical site (putting a significant amount of mental and physical burden on them [37–39]) and raising concerns about the need for extensive training and possible restrictions for conducting a variety of surgical tasks [1,20,40,41]. De facto, indirect manipulation, limited perception, mirrored motion, and degraded hand-eye coordination led to the design of the first teleoperated endoscope, named AESOP (Automated Endoscope System for Optimal Positioning) [42]. Using AESOP, the surgeon, located at a very short distance, could move the laparoscope in every direction in the field of view of the video feedback by giving commands through foot pedals and sensorized hand controllers. The experience gained through the use of AESOP motivated the development of a wide range of assistive robots in the operation rooms resulting in several subsequent technologies, such as the Laparoscopic Assistant Robot System (LARS by IBM, Armonk, NY, USA) [43], which was used to manipulate surgical instruments in delicate surgeries resulting in "Steady Hand" and cooperative control. Later on (1998–2003), Zeus (Computer Motion Inc., Goleta, CA, USA) [44], built upon AESOP by adding new arms manipulating surgical, was used for the first transatlantic telesurgery "Operation Lindbergh" [45]. It was then replaced by the successful da Vinci Surgical System (from Intuitive Surgical Inc., Mountain View, CA, USA) [46], which added the 3D stereoscopic vision (enabling depth perception for the surgeons, which was detrimental in previous telerobotic systems), and enhancing maneuverability of MIS instruments. This technology was initially aimed at minimally invasive abdominal surgery [47], but it rapidly opened to other clinical applications [48]. It can be mentioned that da Vinci Surgical System was a complete revolution in the field of medical robotics. More details are provided in the subsequent sections of this paper.

After the invention of the da Vinci Surgical System, other industries and academic centers proposed new robotic systems for specialized surgeries [49,50]. For instance, in 2009, the German Aerospace Center (DLR) designed the MiroSurge telerobot for experimental research purposes. This robot introduced force feedback: a major missing function in commercial telerobotic systems. Haptic feedback is still an open research question, and several techniques and technologies have been proposed in research centers and industries with the hope of addressing the corresponding challenges (which mainly associate with the cost, instrumentation, transparency, and stability); see Section 2. We can also mention the Flex (Medrobotics Inc., Raynham, MA, USA), specifically designed for use in Trans-Oral Robotic Surgery, the MAZOR SpineAssist/Renaissance® and the ROSA® (Zimmer Biomet-Medtech, Montpellier, France), both dedicated to spinal surgery, and the BrightMatter™ suite (Synaptive Medical Inc., Toronto, ON, Canada) robots [51]. Among the more recent introductions on the international market [52,53], the following products and companies can be mentioned: the Senhance (TransEnterix, Morrisville, NC, USA, approved by FDA for gynecologic and colorectal procedures in 2017), the Mako (from Stryker Corp., Kalamazoo, MI, USA, for total knee and hip replacement), and the Versius Robotic System (Cambridge Medical Robotics, CMR Surgical, Cambridge, UK, approved for use in Europe and currently studied by the American FDA).

*1.4. The Motivations for Telerobotics in Medicine*

The above-mentioned efforts illustrate the high interest in telerobotic systems for revolutionizing the field of medicine. The ultimate goal of teleoperation, especially in medicine, is realizing the perfect "telepresence", a concept popularized in the robotic community by Sheridan in 1995 [54]. Telepresence is the sense of being and acting at a location other than where one is. Even if this may be seen as an unreachable objective in a short time frame (especially due to the complexity of robotized perception), the research and development in this field have been actively followed by many researchers and industries to enhance the existing technology (knowing that robotics-mediated haptics sensation is far from the actual perfect direct touch, in terms of perception) [55].

One of the initial motivations for surgical teleoperation is the need for a long-distance medical operation. At the scale of a country, rural hospitals often lack equipment and experienced surgeons. To investigate the feasibility, in early 2000 some experiments have been performed. In this regard, in 2003, more than 20 telesurgeries were conducted at a distance of 400 km in Ontario, Canada, with a Zeus-TS surgical robot [56]. Two surgeons (one in each hospital) could take the local control of the robot when desired. This successful experience has shown the feasibility and potential to use telerobotic systems for telesurgery in rural and underprivileged areas. At a higher distance, since 2004, the National Aeronautics and Space Administration (NASA) has conducted experiments on remote laparoscopic surgery for astronauts in space conditions, initially with a Zeus robot [57]. The main difficulty, in this case, was a significant time latency. It has been shown that telesurgery performed by an experimented surgeon located on earth was more effective than an on-board surgery conducted by a less expert surgeon on board. Despite these motivating experiments, the current regular practice requires the patient and the surgeon to be in the same hospital (and in most cases the same room). Nonetheless, telesurgery has the potential to facilitate "the exchange of medical expertise around the world without requiring physicians to travel." This saves time, money, and effort by bringing the remote operating room to the fingertips of the surgeon and vice versa [58].

Teleoperation is also required for medical manipulations in workspaces unreachable by the surgeons' hands. For instance, MRI-guided interventional robotics is an emerging field. MRI technology is a widespread high-quality imaging technology that has shown great potential to be used for real-time visualization of deep soft tissues [59]. However, MRI machines require fixation of the body region in a long and small tube which significantly limits the possibility of conducting a manual operation. Due to the electromagnetic nature of this machine many operational tools and other machines cannot be used. In 2012, Seifabadi et al. demonstrated the feasibility of MRI-guided prostate biopsy using an MRI-compatible teleoperated needle driver module based on piezoelectric motors [60]. In 2013, a solution combining pneumatic and piezoelectric actuation was introduced [61]. A recent review on this topic can be found in [62,63].

Besides surgery, there are several other medical applications of telerobotic systems. In this regard, in Section 2.2, we will introduce telerobotic rehabilitation, which is an emerging field of medical telerobotics. Telerobotic rehabilitation has the potential to transform how remote patients with motor disabilities (such as post-stroke ones) can receive kinesthetic motor therapy over a communication network and possibly when they are at home interacting remotely with a clinician in a hospital [7,64]. This technology provides an equal opportunity of accessing rehabilitation services, regardless of geographical limitations, and provides patients with an immersive experience of teletherapy and interpersonal interaction. This paradigm for delivering intensive active-assist therapy attracts more interest recently due to the increased awareness caused by the COVID-19 pandemic. Without such technology, patients who may be considered at high-risk would potentially receive a degraded quality of therapy during a global health crisis (such as a pandemic), due to the limitations around accessibility to public places and requirements regarding reducing the frequency of interpersonal interactions in the healthcare systems. This topic is more discussed in Section 2. It should also be mentioned that the conventional use of a telerobotic

system enables direct supervision of the operator over the conduction of the task. Indeed, more advanced telerobotic systems have embedded local intelligence allowing for the realization of shared-controlled tasks (see Section 3.2), during which the operator and the machine share the autonomy for task conduction. Several examples include telerobotic systems which compensate for the surgeon's hand tremor during operation, scale down the surgeons motor control to enhance the precision, provide forbidden and guiding virtual fixtures, or compensate for the movement of the organs during operation. The shared autonomy frameworks are designed to reduce the mental and physical burden on the surgeon and maximize the quality of surgery up to the level which is not possible to achieve without the use of a telerobotic system. Some examples can be found in [2,65–76].

Another interest of surgical telerobots is their ability to help train surgeons. For instance, in [77], the daVinci robot is able to record and simultaneously playback laparoscopic videos, robot arm motions, and surgeon–console interactions. A user can then replay on-demand recorded procedure sections with the opportunity to watch stereo videos and simultaneously feel the recorded movements on the hand controllers, at adjustable speeds. This approach has been recently used in [78] to determine experimentally how a surgical robotic system can be used for training novice surgeons in conventional laparoscopic. These authors show that combining the playback function and discovery training (trainees learned a surgical task by themselves through trial and error) leads to the best accuracy compared with trainees who used playback or discovery alone. It is valuable to record surgical robot usage by surgeons to collect data for several purposes: (1) training purposes as exposed earlier, (2) the development of advanced control algorithms enhancing the autonomy levels of surgical robots, (3) retrospective error analysis [77]. Yet in 2017, Nagy et al., envisioned that all the objective data available (with patient-privacy constraints) provided by these recordings and surrounding sources (such as medical imaging systems) could be used through big-data analysis for the three aforementioned purposes. For example, surgical data science should therefore "*provide better patient outcomes and a reduction in healthcare costs*". An example of such approach is provided by Wang et al., in [79]. They experimentally showed the efficiency of machine learning to provide automated skill assessment in surgical training using public MIS robotic datasets.

*1.5. The Critical Issue of Stability and Transparency of Telerobots*

One of the main performance metrics for any telerobotic system is the transparency of the coupling. Transparency means that the degree to which the operator would feel the environment and deterioration of transparency would result in deterioration of perception. This can be caused by a violation in velocity tracking at the environment side or force tracking at the operator side. One of the main challenges for transparency is the dynamical behavior of the robots in the loop of teleoperation. To address this challenge initially, Lawrence's four-channel teleoperation (see Figure 1 with $C_5, C_6 = 0$) was introduced in the literature [80], which would require the sending of both force and velocity from one side to the other side. However, due to the complexity of the four-channel design and the low stability margin of the architecture, an extended version of the four-channel design has been proposed by adding a local force control loop on both operator's side and the environment's side [81] (see Figure 1). The extended Lawrence architecture can drop one channel and achieve three-channel teleoperation while guaranteeing transparency (see Figure 1 with $C_3 = 0$). A modification of extended Lawrance architecture has been proposed by Atashzar et al., which allows for achieving perfect transparency only with two communication channels, reducing the complexity of the system and increasing the stability margin (see Figure 1 with $C_3 = C_4 = 0$). Although the transparency of the system can be achieved using the aforementioned architectures, stability has always been a major concern for the designers of telerobotic systems. Stability is one of the main bottlenecks of adding haptic feedback to surgical telerobotic systems. Using hybrid transfer matrix evaluation of a transparent teleoperation system, it can be shown that stability and transparency are two conflicting criteria in telerobotic systems. As a result, any stabilizer would result in

deterioration of the transparency of the system. More advanced stabilizers optimize the amount of waveform deterioration, minimizing transparency degradation. It should also be noted that some features of telerobotic systems such as scaling, can affect the stability of the system since it directly affects the loop gain, and it can be shown that based on the small gain criterion, increasing the feedforward gain of force from the environment side to the operator's side results in reducing the stability margin of the system. This should be considered if force magnification is considered as one of the features of the system.

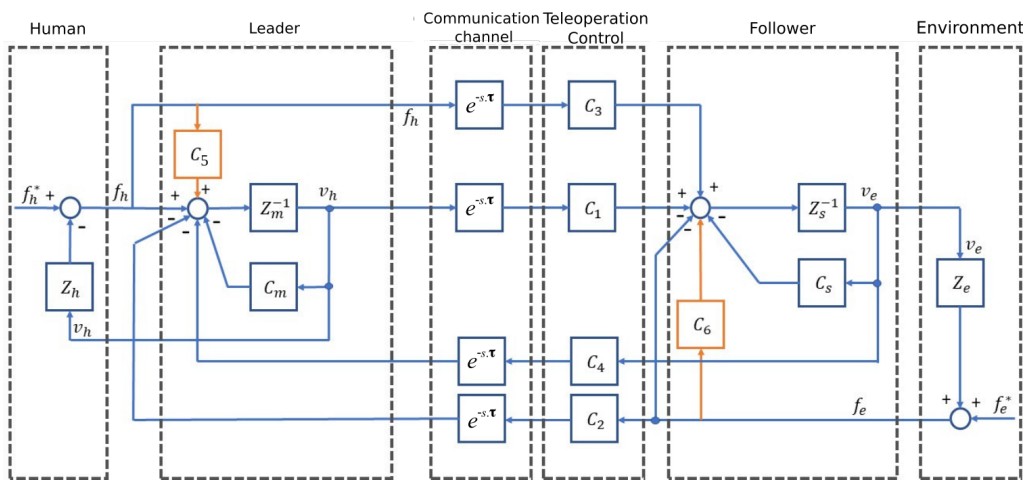

**Figure 1.** Lawrence's four-channel teleoperation (which becomes E-LFC when $C_5, C_6 \neq 0$).

One of the main challenges of haptics-enabled telerobotic systems is the communication delay. It can be mathematically shown that the delay would inject extra energy into the system resulting in accumulation of energy in the closed loop system which can (under some circumstances) result in instability and divergence. There has been a wide range of research and development activities to design and implement various stability frameworks for robotic and haptic systems to deal with the instability caused by the delay. Most of the existing approaches function based on Weak Passivity Control Theory (W-PCT). For this, the environment and the operator are usually considered to be passive, while the communication network is considered to be the source of nonpassivity due to the communication delays. Based on W-PCT, by making the communication passive, a telerobotic system will be converted to a negative interconnection of passive subsystems (Human, Communication, Environment), which will be passive and thus stable. The existing approaches usually provide local (on one side) or distributed (on two sides of the communication) damping to dissipate the extra energy introduced into the system by a nonpassive interconnection link and guarantee global passivity and thus stability. In this regard, the wave variable transformation (WVT) rooted in scattering transformation (ST) is the most commonly used approach in the literature [82] (see Figure 2). Using WVT, the transmitted force and motion signals are converted to wave variables. It has been shown that in the presence of unknown constant communication delay, WVT results in the passivity of the network. An important feature of WVT is that the transformation at the leader side is the inverse of transformation at the follower side. As a result, in the absence of communication delay (which would result in the passivity of the communication), the two transformations would cancel out each other, and thus no signal deviation would be imposed. However, in the presence of communication delays, the two transformations cannot completely cancel out each other, and this would result in a deviation of force and velocity tracking. WVT utilizes a tuning factor named wave impedance, which distributes the deviation from the reference between the force and velocity channels. A major problem with WVT is the sensitivity to variable time delays. Various versions of the WVT have been proposed in the literature (see [83] for a 2014 review), some of which are to make it robust to variability in the time delay, for example, using wave scaling, the intensity of which depends on the variability of time delay (the higher the variability, the lower the scale). Although the problem has been

addressed theoretically, this approach results in excessive deviation of tracking through the need for wave scaling. Wave scaling modifications do not tolerate delays that change abruptly (when derivation of delay is greater than unity).

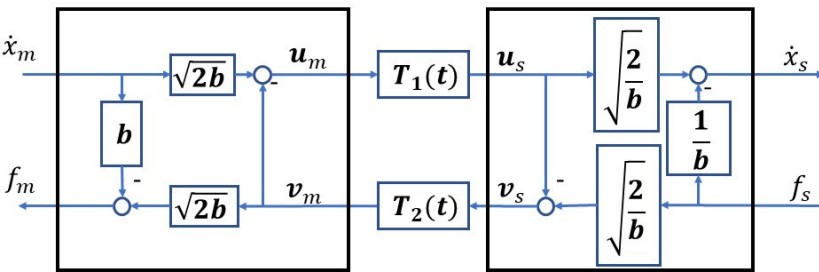

**Figure 2.** Wave variable transformations for communication link passivation.

Another approach for stabilizing the telerobotic system in the presence of variable and unknown time delay, relaxing the assumption of slow changes of delay, is the time-domain passivity control (TDPC) approach [84]. This class of adaptive nonlinear stabilizer utilizes the concept of passivity observer (PO) and passivity controller (PC) (see Figure 3). PO observes the passivity condition of the system by monitoring the energy flow, and PC injects adaptive damping into the system to dissipate the extra energy. This approach was originally implemented in the energy domain; however, a modification of this approach has been implemented in the power domain to result in a smoother control behavior by distributing the dissipation over time. TDPC has been used as a one-port controller when it is placed only at the leader side, and it has also been used as a 2-port controller when two sets of PO/PC are placed on the two sides of the communication network to reduce the conservatism of the system. It should be noted that the one-port design of TDPC makes it possible to use that for the haptics system when the environment is virtualized, and this is another difference between TDPC and WVT. The above-mentioned approaches guarantee the stability of the system, assuming that the environment and the operator behave like inherently passive systems. However, in the context of rehabilitation or telerehabilitation, the behavior of virtual or human therapists would include the injection of mechanical energy into the system to empower the patient and make the task feasible. Energy injection is a non-passive behavior violating the main core assumption of conventional approaches. To address this problem, Atashzar et al. has extended the design of the passivity-based stabilizers and designed a new class of nonlinear adaptive stabilization framework based on strong passivity theory (SPT), which does not require passivity of all included components [7,85]. For this, the excess of the passivity of the user's biomechanics is identified and used in the design of two passivity-based stabilization frameworks of the mentioned family. It has been shown that with the use of this new family of stabilizers, there is no need for assuming passivity on the environment side addressing the problem with telerehabilitation and assistive haptics-enabled systems. In this regard, this new family of nonlinear stabilizers considers the amount of energy that can be damped out by the biomechanics of the user, and this makes a margin of passivity which allows the environment to be non-passive to some extent (depending on the excess of the passivity of the operator). Besides passivity-based stabilizers, Small Gain Control Theory has also been used in the literature not only to analyze the stability behavior of the system but also to synthesize a new class of stabilization. The benefit of using small-gain control is that it does not make any assumption on the passivity behavior of the system; since it cares about the overall loop gain without consideration of the sign of signals. Due to its unique behavior, the Small Gain Control (SGC) approach has been proposed by Atashzar et al. in the context of telerobotic rehabilitation [64].

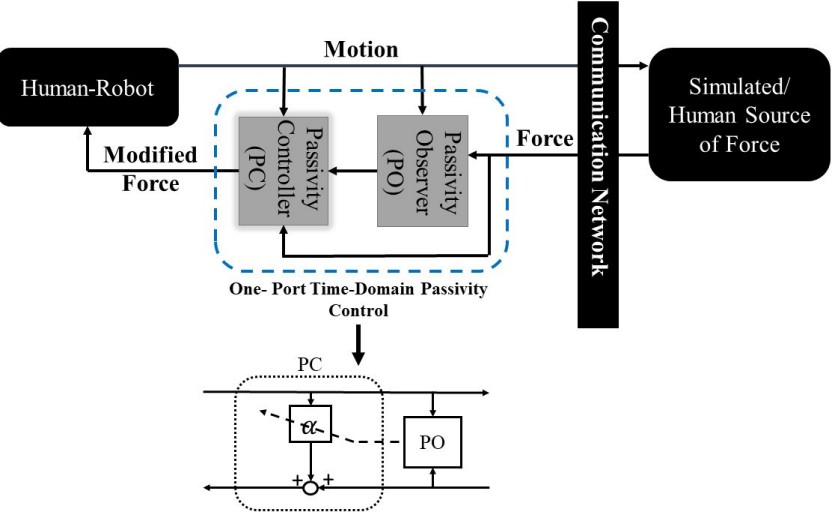

**Figure 3.** Time-domain passivity control (TDPC).

### 1.6. Outline of this Paper

In this paper, we detail how advanced telerobot-mediated medical tasks can be achieved using various topologies of telerobotic systems, beyond conventional single-leader-single-follower architecture. This topic is discussed in details in Section 2. Besides, Section 3 details solutions found in the literature for enabling various levels of telerobotic autonomy.

## 2. Medical Telerobotic Topologies

### 2.1. Single-Leader Single-Follower Topology (SL/SF)

The conventional topology used in telerobotic systems is composed of a single leader robot and a single follower robot. The single-leader–single-follower topology (SL/SF, see Figure 4) has been used, as mentioned before, to augment human sensorimotor competence [28,86]. A successful example of an SL/SF telerobotic system is the da Vinci Surgical System [23], using which it is now possible to conduct minimally invasive laparoscopic surgeries inside a patient's body with an ultra-precision, reliability, and safety [65,87,88]. This topology transfers the pose commands of the surgeon collected by the leader console to the follower robot inserted through small incisions inside the patient body to conduct the surgery under the direct control of the surgeon while augmenting the sensory and motor skills of the surgeons [65,87–90], as explained in the next subsections.

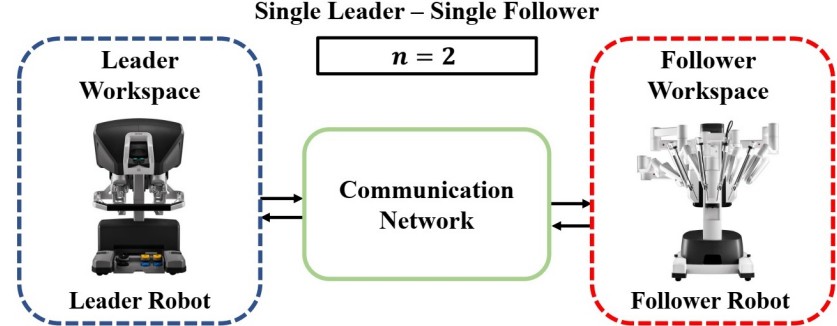

**Figure 4.** Single-leader–single-follower topology where *n* is the number of network ports/terminals (Images ©2020 Intuitive Surgical, Inc., Sunnyvale, CA, USA).

There are several other examples of applications for SL/SF topology in the medical domain (beyond laparoscopic surgery) when interactional forces can be much higher or much lower than laparoscopic surgery. One example is SL/SF telerobotic rehabilitation which allows for remote tele-physical sensorimotor interaction between an in-hospital

clinician and an in-home patient for remote delivery of physical therapy as a new paradigm for delivering intensive active-assist therapy for patients with neurological damages or disorders addressing accessibility issues [64,85,91]. As another example of SL/SF topology telemicromanipulation can be mentioned, for instance, [92] introduces a haptics-enabled SL/SF magnetic micro-manipulation platform with promising potential for a range of micro-scale biomedical applications. The follower robot uses a controlled magnetic field for manipulating a ferromagnetic micro-device when the leader robot is a haptics-enabled device taking position commands of the user (to be scaled down and followed by the follower robot) and rendering the scaled-up force field. On the other side of the spectrum, high-force SL/SF teleoperation has been also used in medicine. Especially in oral surgery [93] and orthopaedics [94–98], drilling is a common high-force and sensitive task. In the rest of this section, we explain in more detail how telerobotic systems can augment the sensorimotor skills of clinicians and enhance the outcome of therapy. We will also discuss the existing challenges and future visions for this topology.

### 2.1.1. Sensory Augmentation through SL/SF Telerobotic Surgery

Telerobotic systems provide the operator with complementary perceptual awareness during the conduction of tasks (such as surgery) to augment the knowledge of the operator (such as the surgeon) regarding the condition under which the operation is being conducted. This augmentation has been achieved through visual [40,99], auditory [100] and haptic [2,35,101,102] channels. Telerobotics-assisted sensory augmentation in surgery is particularly designed to address the three critical sensory restrictions, which exist in manual minimally invasive surgeries, namely (a) degraded hand-eye coordination [103] (b) lack of depth perception [20,104], and (c) insufficient or inaccurate haptic feedback [35]. See [63] for details about desired characteristics of haptic interfaces.

In most commercialized examples, visual augmentation has been the main focus. Through the use of the stereoscopic 3D vision systems at the leader robot console, surgeons can intuitively perceive the depth position information during operation while not needing to wear heavy and fatigues head-mounted displays [20,105]. This was not achievable using conventional hand-held minimally invasive systems [65,87]. Since the forward and inverse kinematics are solved locally, and the two robots are synced in task space, the conventional issues such as mirroring motions (which exist in manual minimally invasive surgery (MIS)) are compensated for [2,106–109]. The line of sight of vision directly collides with the hand motion frame; thus, the hand-eye coordination and visuomotor misalignments are all corrected when compared with manual MIS [10]. This has made the visuomotor control of the leader-follower telerobotic surgical system a superior compared with conventional manual MIS. Besides, surgeons have direct access to more advanced methods of augmentation, such as using fluorescent cameras in the same visual system, which allows surgeons to detect pathological tissues such as affected lymph nodes [110]. Preoperative images have been used to visually guide the surgeon during the surgeries for enhanced accuracy [67,111,112]. However, one has to take into account its limitations: in 2016, Meccariello et al. stated that the use of visual haptic feedback (instead of real force) did help surgeons compensate for this lack [113], but also increased the average applied force magnitude on the tissue by 50%, and the peak applied force by at least a factor of 2, while the introduction of real force feedback decreased accidental tissue damage by a factor of 3 [114].

Besides vision augmentation in laparoscopic surgeries, advances in multimodal robotics also contributed to new image-guided percutaneous procedures [115,116] including robotic catheterization, when recurrent X-rays imaging has helped the clinicians during navigation. In this regard, through remote manipulation of the catheter, health professionals are no longer exposed to X-rays and can navigate the medical probe using a telerobotic medium. Other benefits are better ergonomics at work for the clinician, besides higher precision of outcome, and shorter operational time (which reduces the radiation to the

patient). More recent image-guided telerobotic surgical systems for vision augmentation include MRI-compatible robots and MRI-based robot navigation for MIS [117].

Deployment of visual augmentation has been successful using telerobotic surgical systems, and commercialized examples have been using mentioned advanced features [1]. Besides visual augmentation, auditory cues have also been provided for the surgeons to enhance sensory awareness [118–120]. However, since the audio channel has been occupied heavily in operating rooms for communication between surgical team members, in commercialized examples, the auditory channel has not been utilized extensively. This kind of feedback has nevertheless interesting applications in basic surgical training contexts, as demonstrated in [121].

Regarding haptics-based augmentation it can be mentioned that despite two decades of research and existing evidence of a significant benefit of haptic feedback in surgical operations [100,122,123], the commercialized surgical robotics systems are not equipped with haptic feedback [35,36,55,101,109,122,124]. The reason for this absence is explained later in this section. However, before investigating the corresponding reasons, the importance of haptic feedback in surgery should be discussed, as given in the following. In the literature, it has been shown that haptic sensation provides critical information during manual surgery regarding (a) the mechanical characteristics of tissue (such as texture, stiffness, size, and location of tumors) [125], (b) the amount of force which may cause damage on the tissue [122,126,127], and (c) quality of tool manipulation to avoid issues such as suture breakage and needle slippage [40,99]. It is also known that haptic information is very critical for reducing surgical errors [35,36], and tissue damage [122,126,127], besides operating times [1]. The mentioned knowledge has been achieved by systematic studies conducted in many cases on manual surgeries due to the conventional concerns related to degraded haptic sensation in manual MIS caused by the following factors:

- friction inside the cable-driven tools [128–134], also between the tool and the trocar [125,135],
- the disturbing forces at the trocar and abdominal wall, the disturbing forces on the tool by nearby organs [136],
- the small point of contact between the tool and tissue causing low signal to noise ratio, and the long mechanical distance between the point of contact with tissue and the hand of the surgeons [36,136,137].

Despite the benefits of minimally invasive surgeries, such as reduced blood loss [30], reduced damage to the tissues [31], reduced risk of infection [32], reduced recovery time [33], and cosmetic benefits [34], the limited sensory awareness (caused by factors mentioned in the above) has been a conventional bottleneck for manual MIS [35,36]. The mentioned challenge has called for extensive and specialized sensorimotor training for manual MIS surgeons and has raised excessive concerns about the ability of surgeons to perceive the status of the surgical site even for experienced surgeons [1,20,40,41]. Thus, in many complex surgical cases, MIS is known to be excessively challenging for surgeons to conduct, putting a significant amount of mental and physical burden on them [37–39]. The situation is more challenging for microsurgeries, as the amount of force is much lower than the ability of human perception, which has made those surgeries visually heavy and cognitively complex [66,138–144].

However, the above-mentioned restrictions on haptic sensation can be addressed using robotic surgical systems. In theory, small arrays of tactile sensors can be mounted on surgical tools which can detect the interactional forces during the surgery [7,122,125,136,145,146]. The measured force signals can then be filtered, processed, amplified, and sent to the force control loop of the actuated leader robot, which can then replicate the resulting force for the surgeon, increasing the haptic awareness during surgeries [101,102,147]. It should also be noted that the measured force can also be visualized and sonified for the surgeon indirectly increasing the haptic awareness in the absence of direct feedback. This technique is known in the literature as a sensory substitution [40,99,100,145,146] and is usually suggested as

a replacement for direct force feedback when the stability of closed-loop force feedback cannot be guaranteed. Besides, to address the issue of the small amplitude of force during micro-robotic surgery, such as retinal surgery, ultra-small, and accurate force sensors have been suggested in the literature which can measure micro- to milli-Newton forces, and be magnified for the surgeons for better awareness [66,138,139]. The above-mentioned topic has been studied in the literature under the umbrella of haptics enabled telerobotic surgery, and Longmore et al. recently listed direct and indirect force-sensing solutions in [52].

The benefit of adding direct or indirect haptic feedback to robotic surgical systems has been studied extensively in the corresponding literature, some of which were conducted on the da Vinci surgical robotic system [100,122,123]. Clinical benefits have been reported such as (a) reduced average and peak forces on the tissue, and tissue damage [122,126,127]; (b) enhanced quality of knot tying, reduced frequency of suture breakage during surgery, enhanced force consistency [40,99,148], (c) realization of haptics-enabled tumor localization [125], and improved safety (for the patient) and lowered fatigue (for the surgeon) [148].

However, haptic feedback has not been achieved yet in commercialized examples [55,101], and this is known to be one of the most significant limitations of existing telerobotic surgical technologies motivating researchers and industries to locate the challenges and invest in mitigation strategies [108,109]. Still in 2017, there was no consensus about the best way to provide haptic feedback to the user, and also, no human factors and ergonomic (HFE) was available to prove the interest of each in practice [149]. There are two major challenges in this regard, namely: (a) instrumentation [136], (b) stability [7,82,85,150].

Regarding instrumentation, it should be noted that sensors to be used on robotic surgical systems should be miniaturized, sterilizable, biocompatible, durable (to pass sterilization process), disposable, and cost-effective (since they need to be disposed of after a few uses to avoid infection transfer between patients and warranting safety) [36,125,136,151]. The amplitude of the force signal is relatively high for laparoscopic surgeries and is very low for microsurgeries. Implementing a sensor that can measure multidimensional forces while providing a high signal to noise ratio and meeting all the above-mentioned requirements is a technically challenging task. There has been significant research on this topic, and as one solution (for some applications), optical force sensing, in particular fiber Bragg grating (FBG) sensor, is introduced as the technical solutions for this challenge [152–155]. FBG sensors allow keeping expensive, unsterilizable, and technologically-challenging components outside of the patient's body while only some thin fiber-optics are connected to the tool, occupying a very small volume in space and measuring forces with high accuracy. FBG sensors are flexible and can be mounted on the body of the surgical tools, or inside the tools, or at the tip of the tool to measure different force features. Besides, the optical fibers are inexpensive, disposable, and biocompatible [153–155]. As a result, optical force sensing is known to be a future direction for enabling the next generation of telerobotic surgical systems in reflecting force sensation [154,156–162].

However, even if the forces are measured accurately, it is not trivial to reflect the force for the surgeon while guaranteeing stability [67,111,163–165]. It has been shown in the literature that high transparency of force reflection by leader robots and stability of telerobots are two opposing criteria [3,64,85,166–169]. Thus, a haptics-enabled telerobotic surgical system that has near-to-ideal transparency in force reflection may suffer from poor stability and vice versa. The poor stability would sacrifice the safety of physical interaction between the patient and the robot, which is absolutely not acceptable in particular when robots interact with humans [1]. There are conventional stabilizers, such as wave variables, or time-domain passivity approaches, which degrade the transparency to guarantee the stability of the system. It should be noted that, due to the neurophysiology of haptics sensation, distracted force feedback may sometimes be even worse than not having the feedback at all. This is because humans rely heavily on haptics modality for conducting delicate tasks; thus, high quality of force reflection can significantly improve the performance of fine motor control of the surgeons using the robotic surgical system [146,170,171] while a low quality of force feedback may potentially defeat the purpose. Degraded force feedback

can degrade the performance by misleading motor control of surgeons, when compared with the lack of force feedback, which would push surgeons to put all the weights of sensory awareness on to the visual channel [145,146,170–173] which in many cases has been considered as a successful practice through visuohaptic skill development.

It should also be noted that the instability challenge will be magnified significantly if the leader and follower robotic systems are placed in two distant locations requiring a significant networked connection, which would result in variable time delays with significant jitter, data loss, and latency [172,174–177]. With the invention of 5G and beyond networked systems and major promise in maximizing the reliability of data transfer while minimizing the time delay, it is now possible to think about solutions that can not only address the stability [178,179] of local leader-follower telerobotic surgical systems but also networked telerobotic surgical systems. For this purpose, advanced control techniques have been recently proposed based on small gain control theory [64,180] and strong passivity control theory [7,85,166,181], which can potentially rely on an ultra-reliable 5G communication network to guarantee the stability and to maximize the performance of force reflection. There are several different studies on the amount of allowable delay in a telerobotic system, especially when used for surgical operation. Various numbers are given in the literature, mainly in the range of 100 to 200 ms. However, it should be noted that the amount of delay depends on many different aspects. For example, the type of operation and the sensitivity of the surgical submissions can significantly affect the allowable delay. For example, if the surgeon is operating near to a beating heart, the acceptable delay would be very low compared to some other surgeries when the organ is less sensitive or when it does not have motion. This is about the feedforward path for position control at the follower side. It should be noted that in the case of force-feedback teleoperation, the delay can (a) affect the perception of the stiffness of the environment, and (b) can destabilize the system resulting in safety concerns which should be stabilized by adding controllers which can then further affect the fidelity of force feedback. Because of all the reasons mentioned above, it is not possible to provide a fixed number for the acceptable delay in a medical telerobotic system. Readers will find more recent information on this topic in [27,182,183].

Thus with the use of advanced technology, including ultra-fast communication, advanced instrumentation, and algorithmic stabilizers [7,64,82,85,150,166,180,181,184], it can be envisioned that the next generation of telerobotic surgical systems is equipped with means of high-fidelity haptics reflection enabling surgeons to not only benefit from an augmented vision but also augmented haptic sensory feedback.

### 2.1.2. Motor Augmentation through SL/SF Telerobotic Surgery

Besides sensory augmentation, telerobotic surgical systems are known to augment the motor performance of surgeons to directly correct and enhance the manipulations generated by the surgeon to minimize errors and increase the quality of surgery. In this regard, tremor compensation [2,65,66], organ motion compensation [75,76], surgeon's motion scaling [2,5,185], guiding force fields [67–71], and forbidden virtual fixtures [67,68,111,112,186–188] are existing examples of motor augmentation achieved using teleoperated robotic systems [1].

Regarding tremor compensation [65,87,189], it can be mentioned that through basic low pass filtering or in a more advanced manner through accurate estimation of hand tremor in realtime with minimum phase lag using advanced signal processing modules, such as band-limited multiple Fourier linear combiner [190–193], it is now possible to extract and predict the high-frequency, low amplitude, involuntary tremorous motions of the surgeon's hand and damp them out before sending the motion to the surgical robot [1,189]. Thus, through the computerized signal intervention achieved by leader-follower telerobotic systems, the tremorous motion of the surgeons can be converted to smooth ultra-fine motor commands, which increases the quality of delicate surgeries (in particular for microsurgeries such as in retinal operations) [66,138,139].

In addition to tremor compensation, using leader-follower teleoperated robotic systems, it is possible to scale down the motions generated by the surgeon [1,2,5,185,194]. This would result in converting a very fine motor requirement to a course motor task for the surgeon. This would significantly increase the accuracy by reducing the complexity of control [5]. For example, the surgeon can move the leader robot accurately in centimeter ranges, while the motion can be scaled down to micrometer and can be replicated by micro-actuator in delicate surgeries. This framework is also valid for larger-scale operation such as laparoscopic operation when the surgeon can decide to scale down the motion to reduce the burden of fine motor control using telerobotic systems [194]. In the literature, motion scaling and tremor filtering are identified as two critical features of telerobotic surgical systems that enhance the dexterity and performance of surgeons when compared with manual surgeries [2,5,185].

Regarding organ motion compensation, it can be mentioned that through advanced computer vision and image processing modules (most of which have a Kalman filter core algorithm [11,192,193,195–198]) it is now possible to measure motions of moving organs such as heart and lung during the operation and compensate automatically for the physiological motions while the surgeon can be responsible only for providing relative positional commands. The image modalities used for these applications are usually acquired by endoscopic cameras or ultrasound probes connected to the robotic tools [10–13]. A major challenge for this motor augmentation task is the large latency for processing the images when compared with fast organ motions [11,199–201]. For example, generically, heart motion during surgery (after deceleration by medication) can be as high as 210 mm/s with an acceleration of 3800 mm/s$^2$ [199]. For addressing this issue advance and lightweight signal processing modules such as Kalman filter [11] and advanced control modules such as model predictive control [202] have been suggested in the literature and sometimes are fused with other modalities such as electrocardiograms (ECG) to track, predict and compensate for the phase lag [11,199–205]. This task is a form of shared autonomy (see Section 3.2) using which robots can utilize computer vision to compensate for the repetitive motions [12,199,203,206–208]. This will significantly increase the accuracy of the operation since it significantly reduces the mental, cognitive, and physical load on the surgeons during operation [10,11,200,209–212].

Regarding the virtual fixture concept, it can be mentioned that leader-follower robotic systems can provide kinesthetic corrections, guidance, and avoidance for the surgeons based on the fusion of preoperative and intraoperative information [1,67,68,111,112]. Virtual fixtures (also known as "active constraints") are high-level algorithms that generate a kinesthetic no-fly-zone for telerobotic systems. For this, the algorithm would generate a virtual spatial manifold (the topology of which depends on the task) with specific stiffness (usually high stiffness). As a result, when the user hits the manifold in the space he/she would feel a resistive/repulsive force avoiding penetration to the virtual "wall". Although typically virtual fixtures do not suffer from communication delays, they still are virtually rendered, thus the stability of interaction would be challenged by the digitization (the sampling period). This challenge is pronounced since usually virtual fixtures have very high stiffness which would significantly reduce the margin of stability making the system susceptible to even the smallest delay caused by digitization. This concept has been discussed in the literature under the topic of Z-width which shows that each haptic system can safely display a limited range of impedances (this range is called Z-width). In 2014, Bowyer et al., reviewed the existing algorithms found in the literature and detailed the various strategies to overcome this issue [67]. As mentioned, one main issue is reproducing a haptic interaction with highly rigid virtual walls. As mentioned, it has been shown that the sampled nature of the control loop can virtually generate non-passive energy and destabilize the whole control loop [84]. Recent works on this issue propose using the H∞ approach to design controllers capable of rendering stiff walls (<5000 N/m, much higher than with other control approaches) in an oscillation free manner and with actuator force limitation to avoid their saturation. For more viscous interactions such as the ones

encountered in neurology, recent works are brought by Gil et al., in [213]. The forbidden region virtual fixtures are to restrict surgeons' operation on sensitive tissues (such as major arteries or delicate organs) through the generation of repulsive force field provided kinesthetically for the surgeon and through stopping the follower robot from hitting those no-fly zones [67,111,112]. This feature directly enhances the performance, accuracy, and safety of the operation by correcting the dangerous maneuvers by surgeons, (in particular, novice surgeons) and preventing damages [68,111]. Using repulsive virtual fixtures, the telerobotic system can continuously monitor the motion of the surgeon for some forbidden regions [69,214]. This technology has been very successful for operations on rigid tissues (such as bones) or structured organs such as retinas [68,186–188,215], but it faces significant technical difficulties for soft, highly dynamic, and unstructured soft tissues due to technical difficulties in registering preoperative and intraoperative information sources. There is active research on this topic under the umbrella of Dynamic Active Constraints (DACs) [123,151] for realization on soft tissues.

In an opposite scenario, using guiding virtual fixture telerobotic systems can assign novice surgeons to converge to a desirable trajectory or path (which can be annotated preoperatively by expert surgeons) [68–71,118,216,217]. This technology is usually used under training environments (not during actual surgeries) when novice trainee surgeons practice surgery with robots on highly structured and clean surgical training phantoms mimicking an actual surgical site and allowing the implementation of guiding virtual fixtures for helping trainees with learning and developing a motor task [136,163,206,208,218].

### 2.2. Single-Leader Single-Follower Telerobotic Rehabilitation

During the last decade, the topic of robotic rehabilitation has attracted a great deal of interest as adjunct (or ultimately replacement) interventions, which can significantly reduce the load on healthcare systems (please see [219]).This is since stroke is the leading cause of significant motor disabilities and results in excessive economic pressure on healthcare systems.

Telerobotic rehabilitation [64,85,91] is a natural extension of robotic rehabilitation, which can provide equal opportunity, regarding access to rehabilitation services, to the people, regardless of geographical and accessibility limitations. Telerobotic rehabilitation architectures allow for remote multimodal and tele-physical sensorimotor interaction between an in-hospital clinician and an in-home patient. The technology has been recently proposed, and several research centers are focusing on the realization of such a technology. Telerobotic rehabilitation can provide patients with an immersive experience of real-time teletherapy and interpersonal interaction. The teleoperated system realizes a new paradigm for delivering intensive active-assist therapy for patients regardless of accessibility issues.

This technology has not been utilized on a large scale yet. However, due to the COVID-19 crisis, the need for such technology is pronounced. The pandemic has affected the accessibility of those in need of rehabilitation centers, and this is a major concern for patients in isolation and those with co-morbidity resulting in a significant pause for delivery of rehabilitation services. The pause, unfortunately, can have a permanent impact on the lost sensorimotor capabilities of those patients since the chances of post-stroke recovery are at a maximum during the three months following a stroke when the brain has maximum plasticity, after which plasticity is rapidly lost. Telerobotic rehabilitation will address this need by enabling clinicians to have wide-range access to patients across the country (including in rural areas) to conduct various objective sensorimotor assessments and rehabilitation interventions. This is to promote high and equal access to healthcare services and is a major need globally. Beyond accessibility at the time of crisis or for remote areas in the country, the technology can significantly increase the number of hours in which a remote patient can receive rehabilitation and sensorimotor assessment services. More information about the COVID-19 and medical robotics is available in [220–222].

The realization of telerobotic rehabilitation was not possible in the past for reasons such as the requirement of critical sensitivity of active-assist technologies and multimodal

sensorimotor rendering systems to the quality of service (QoS) of communication networks, concerns about reliability and resiliency of the network, and security of data transfers (see, for instance [223,224]). It is shown that latency, jitter, and packet loss not only can deteriorate the fidelity of information rendering, but it can also result in a phenomenon called "non-passive network coupling." This results in an exponential energy accumulation at ports of the telerobotic communication and can potentially result in asynchronous out-of-control behavior of the coupled robotic systems. When the two multimodal robotic modules (e.g., one at the patient's side and the other at the therapist's side) are coupled through a network, degraded QoS can even result in "instability" which can be a significant safety concern and has been a bottleneck for the realization of telerobotic rehabilitation. This calls for the design and implementation of novel control architectures to provide safe remote sensorimotor rehabilitation as a line of development for modern healthcare systems addressing the critical challenges of the current robotic and digital rehabilitation systems. Some recent efforts in this regard can be seen in [7,64,85] in which novel passivity-based and small-gain-based stabilizers are proposed to address the stability issue while maximizing the performance and transparency of force-motion coupling between the remote therapist and the patients.

### *2.3. Multilateral Teleoperation*

A natural extension of the conventional use of bilateral telerobotic systems is multilateral telerobotic systems (see [4] and references therein). Multilateral teleoperation is an upgraded version of conventional bilateral SL/SF teleoperation systems, which consists of more than a single set of leader-follower robotic modules. Multilateral telerobotic systems have at least three terminals which can be connected to single-port modules interacting with a remote task or a remote user. Technically, these systems can be composed of multiple leader robotic modules and multiple follower robotic modules.

According to the topological structure of multilateral systems, they are categorized as Single-leader/Multi-follower (SL/MF), see for example [4,225–229], Multi-leader/Single-follower (ML/SF), see for example [230–232], and Multi-leader/Multi-follower (ML/MF), see for example [4,233–246]. These systems allow for the realization of a variety of new tasks that require multiport communication between distributed modules such as collaboration and interactions between multiple network terminals, multiple robots, and multiple operators enhancing efficacy, precision/accuracy, dexterity/manipulability, loading capacity (through distributed power) and handling capability through joint task conduction and shared autonomy, see for example [218,227,230,247–252].

An example for medical application is tele-mirror-rehabilitation when a human therapist holding the first robot works with a remote patient holding the second robot with her/his left hand and the third robot with her/his right hand (in a mirrored fashion) to provide kinesthetic mirror rehabilitation exercises [247,253,254]. As another example, the same structure can be used for an expert-in-the-loop (EIL) surgical training mechanism to train a novice surgeon on how to conduct tasks such as telesurgery [206,208].

In this section, we will provide a summary of applications with a focus on the surgical domain, functionality, and challenges for teleoperation systems considering trilateral and eventually multilateral topologies. It should be noted that the trilateral topology is a special case of multilateral topology, but because of the extensive work that has been conducted for this particular category, a separate sub-category is considered.

### 2.3.1. Multi-Leader/Single-Follower (ML/SF)

Multi-leader–single-follower (ML/SF) system grants the ability for multiple operators to command one single follower robot through manipulation of their associated leader robot and fusion of their motion command to control the follower robot [218,230–232,255–257]. Applications of these systems can be seen in various forms such as EIL multi-robot rehabilitation [4,247,253,254,258–267], collaborative surgery [163,206,237,268] and surgical training [4,163,206,208,258,260–268]. More specifically, using this topology, multiple sur-

geons can share the control of a single follower robotic system (details of medical autonomy are given in Section 3.2 of this paper). A simple case is through a switching mechanism at the follower position controller so that the two surgeons can take turns to conduct the surgery when the other surgeon would go to an observation mode. This has been realized using the dual-user da Vinci surgical system when an expert surgeon conducts a task and then observe the conduction of the same task by the novice trainee [218,269–274]. Instead of switching, fusion scenarios have been conducted when the motions of the two surgeons are fused through a mixing matrix to control the follower robot. This would allow reducing the errors and increasing accuracy [247,253,258,266,269,270]. The mixing matrix is usually calculated and set based on the expertise level of the surgeon before the operation. In a more advanced manner, it is suggested that the performance of the two surgeons can be concurrently calculated, and the authority level of the surgeons can be shared based on the corresponding performance matrix, which can change and evolve [258,275,276]. The performance measure can be calculated on the fly based on metrics like the smoothness of motion, task completion time, the economy of motion, gaze stability [150,207,247]. In the literature, techniques such as fuzzy logic algorithms have also been utilized to evaluate the performance of the two surgeons in real-time and adaptively calculate the authority level to be assigned [240,241,258,277]. It should also be noted that during a motor training scenario, the motion of the skilled user (such as the surgical trainer) has been seen as a reference for evaluating the performance of the second user, taking advantage of the multi-user design [216,217]. Besides, the expert trajectory has been used to generate a guiding force field at the trainee's robot so that the trainee can develop the needed spatial and visuospatial mapping based on the kinesthetic guidance from the expert side. This is called robot-mediated hand-over-hand training [64,206,208,258,278].

Another use of this topology is to enhance the operability of the task-side (follower) robotic system when there exists kinematic and structural asymmetry, causing discrepancies between degrees of freedom of leader and follower robots. In this scenario, each user may take control of parts of the follower robotic system hence increasing the human interface dexterity [279–282]. As another application, surgical tasks such as needle insertion can benefit from this topology. For this, the needle insertion can be divided into two independent tasks of (1) controlling the position and orientation of the needle, and (2) controlling the depth of insertion [230]. Yet another example of such a scenario is therapist-in-the-loop (TIL) mirror robotic rehabilitation for post-stroke patients, using which the therapist's trajectories (first leader robot) is modulated by the user's hand on the less affected side in a mirrored fashion (second leader robot) to rehabilitate the hand on the affected side (the follower robot). This architecture not only allows for higher safety and reliability of the therapy but can potentially allow for stimulating neural recovery through the medium of mirror rehabilitation [247,253,254].

Motivated by the significant benefits of ML/SF topology in several real-life applications, a variant of this topology is also investigated in the literature when the interaction between the follower robot and the environment is virtualized so that multiple users at multiple leader sides can share the control of a certain task in a virtual reality environment [250,255,256,283–286]. The aforementioned application grants significant benefits for the simulation-based medical applications such as surgical training [287], and non-medical applications such as collaborative sculpting and even online computer gaming [250,255,256]. For virtualized ML/SF topology, the virtual reality environment can be shared in a single server (client–server) format or distributed among users (peer-to-peer) [283].

Despite several benefits, there are some existing challenges with the use of this topology. It should be noted that in both virtualized and real-world usages of the ML/SF topology, a question is the fusion methodology between the commands of different users and the corresponding synchrony [218,284]. It should also be noted that such topology may suffer from a nonhomogeneous distribution of delay between different operators and between them and the follower robot, which calls for a specialized stabilization

framework. One of the major challenges with the distributed delay is the complexity of the stabilization architecture [4,218,272,273]. Improving the performance, stability, and functionality of ML/SF topology is ongoing research, and several advanced topics have been investigated during the last decade. Examples are wave-variable controller [284], graph theory, and network topology optimization [256], feedback-based synchronization controller consisting of a linear compensator with a Smith predictor [288], virtual coupling scheme [285,289], event ordering via global time coordinate frame [290] and remote dynamic proxies [291–295].

### 2.3.2. Single-Leader/Multi-Follower (SL/MF)

Another topology of the multilateral teleoperation system is Single-leader/Multi-follower (SL/MF) structure. In this type, more than one follower is in the loop to improve dexterity, manipulability, performance, and functionality of the system. This topology is usually used when one leader is sufficient for commanding the task, while one follower cannot overcome the complexity and geometry of the task [4]. Examples of applications for this topology are micro-tweezing [227], handling heavy and bulky objects [225], cooperative manipulation [225,226], motion coordination of multiple robotic agents (swarm robotics) [228], assembly line tasks such as for bolt-nut pairs [225]. The main challenges are the synchrony between multiple follower robots operating in a shared environment while ensuring a secure grasp under the command of the leader robot [226]. A control algorithm that has been used to ensure the security of the grasp is a passivity-based control framework for cooperative robotic systems at the follower side [226,227], using which the dynamics of the follower robots are divided into two subsystems; shaping system and locking system. The shaping system gets controlled by disturbance cancellation to make sure the grasp is secured, while the locking system is controlled based on the operator's commands with higher dynamics. This control approach was applied to the bio-operation micro-tweezing task. There are other control strategies designed for enhancing the performance of SL/MF topology such as task-oriented controller [225] that requires knowledge of the task before the operation, and wave-variable controller [229], which is used to synchronize follower consoles in the presence of time-varying communication delays while guaranteeing stability.

### 2.3.3. Multi-Leader–Multi-Follower (ML/MF)

Multi-leader–multi-follower (ML/MF) systems are the most complex teleoperation topology in terms of the number of leader and follower components involved in a task (see Figure 5). To provide an illustration of the kind of robots that enter this category, the da Vinci system is a decoupled ML/MF robot: the operator manipulates independently two handles that each controls one robotic arm. This topology takes advantage of (a) improved dexterity, grasp, and manipulation due to the use of multiple follower robots, and (b) more flexibility and cooperation due to the involvement of multiple operators commanding and monitoring the operation [249]. Such systems can make the execution of more complex tasks possible [296]. Potential other applications include the cooperation between several surgeons with more than two tools, with operators potentially being distant from each other and from the patient. An important feature needed for this topology is for avoiding collision between robots on the side of follower consoles since the control is shared between multiple operators [297]. The challenges for this topology can be categorized into two branches, (a) safety [233–236], and (b) motion synchronization of the systems [240–243]. For the safety of the system, different measures have been taken into account in the literature to avoid collision of the follower robots [233–236]. One approach is through predictive models, which predict the trajectory overtime for the motion of the followers and uses it to ensure safety and avoid collision [233]. Another approach is to predict and render a model of the environment on the leader site instead of transmission of force or velocity signals, which can also improve the bandwidth of the overall system [234,235]. The aforementioned

controllers are mostly known as model-mediated controllers, which use the information about the environment to avoid collisions [233–237].

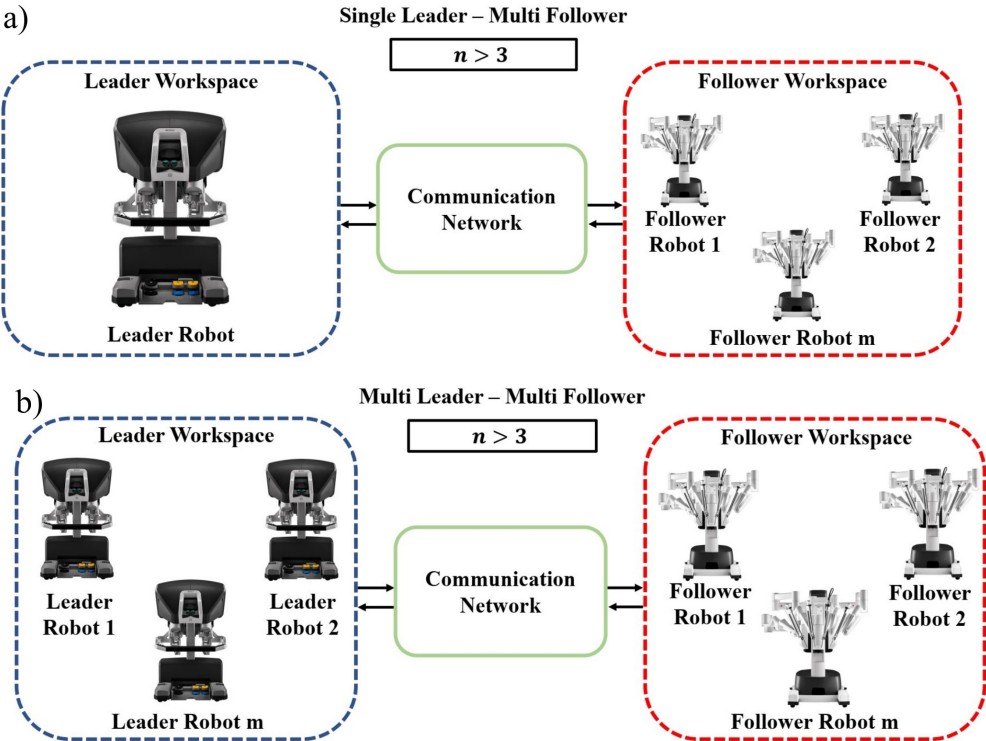

**Figure 5.** Multilateral topologies used in telerobotics, *n* is the number of network ports/terminals. (**a**) Single leader–multi follower (**b**) multi leader–multi follower (Images ©2020 Intuitive Surgical, Inc.).

In addition to motion synchronization and stability, force reflection strategy is another studied topic for this topology. In some scenarios, each operator controls a separate follower robotic arm, while force reflection provides information about the collision. In some other scenarios, environmental force is also reflected. It should also be highlighted that control, local autonomy, and task sharing for this architecture are challenging in the presence of distributed communication delay and uncertainties at each leader or follower side. Several advanced architectures have been introduced in the literature to address some of the mentioned issues, including distributed event-based controllers [238,239], adaptive neural/fuzzy controllers [240–242], and Passivity-based approaches [243–246,249,296,298].

### 2.3.4. Trilateral Teleoperation

Trilateral teleoperation is a specific case of multilateral systems in which the system has three terminals between which the motion and force profiles get exchanged (see Figure 6). Due to the particular use of this architecture, here, we have separately discussed trilateral teleoperation. This architecture has been used in three different configurations, as explained below.

#### Human–Machine Shared Control (HMSC)

This configuration consists of one operator/leader robot, one autonomous agent (please see Section 3 for more details), and one follower robot [206,216,251,299–303]. The configuration has also been seen as a variation of SL/SF architecture, which is augmented with artificial intelligence. In this setting, the operator's actions and the task are monitored by the autonomous agent, which refines the leader's command that is to be given to the follower robot [206,216,299]. Among the scenarios that this configuration has been suggested for, are surgical training and supervised autonomous surgery. For motor

training, the trainee (at the leader side) shares the control of the teleoperated task with the autonomous agent. Haptics cues generated by the autonomous agent can be provided to guide gently or to impose a trajectory strictly to be followed [206,216,299]. For surgical autonomy, the autonomous agent is in charge of monitoring and conducting certain aspects of a task in collaboration by (and under the supervision of) the surgeon, to help reduce the difficulty of repetitive tasks for a surgeon [251,304] (Please see Section 3).

In non-medical scenarios, HMSC systems have been used in telemanipulation, maintenance, and repair procedures for space applications (such as space robots) to take advantage of the intelligence of an operator for decision making and teleoperation, as well as autonomous control of the agent in the loop [300–303].

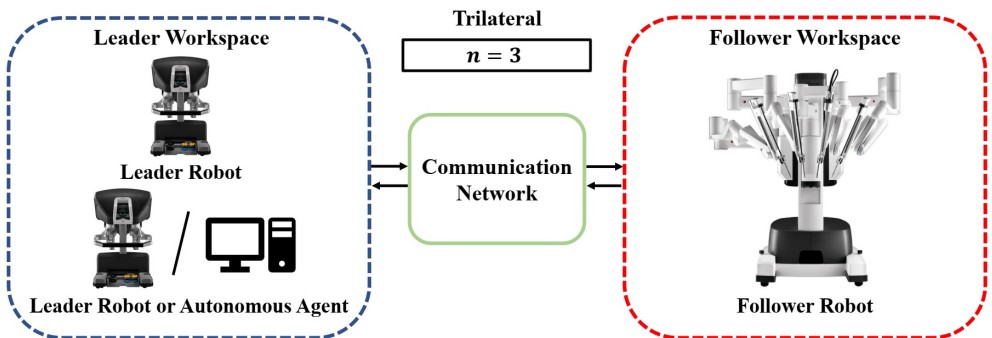

**Figure 6.** Trilateral topologies used in telerobotics, *n* is the number of network ports/terminals (Images ©2020 Intuitive Surgical, Inc.).

Dual-User Shared Teleoperation (DUST)

This configuration refers to structures that have two leader stations for the cooperation of two operators on one follower console. This collaboration helps improve the task execution in comparison with the same task done individually [273]. The concept of haptic-enabled negotiation between two operators can be made feasible using this framework to control the follower console collaboratively through haptic communication [305]. More details on this can be found in Section 3.2 where we compare various autonomy levels. The wide literature related to DUST architectures can be categorized into two main branches of research: (1) control [150,207,269–274,277,306–309], and (2) Functionality [4,247,253,254,258–267].

For controlling DUST topology, subjects such as closed-loop stability have been examined in detail, and researchers have proposed several techniques to overcome existing issues including instability under shared telemanipulation [4,163,258]. In this regard, controllers are developed based on what is known to be the "dominance factor", which defines the level of authority of each leader over the follower action [268]. Several controllers and stability analysis techniques have been introduced for constant dominance factors in DUST structure based on algorithms such as fuzzy controllers [277], basic PID controllers with dissipative gains [308], and passivity control theory [150,207,307].

The concept of variable dominance factor was first introduced by Shahbazi et al. in [163] to enable time-varying real-time adjustment of the authority levels of the two operators. Thereafter, based on methods such as passivity-based approach, and small-gain control, other control frameworks have been examined to maintain the stability of DUST topology in the presence of communication delays [150,207,308,309].

Regarding various functionality of DUST topology, several applications have been introduced in the literature related to robotic rehabilitation and haptics-enabled surgical training [4,247,253,254,258,260–267]. In the context of rehabilitation, DUST topology is used as a way to conduct therapist-in-the-loop (TIL) mirror rehabilitation therapy in which the therapist and the less-affected hand of the patient are controlling the two leader robots, and the more-affected side is connected to the follower robot. With this architecture, not

only the therapist can provide rehabilitation exercises, but also the patient can control the administration of the therapy (considering the comfort, pain, and other factors) using the second leader device. This architecture would generate a mirrored motion between the more affected and less affected sides under the supervision of the therapist and is classified under a mirror rehabilitation framework, which is a well-known technique for promoting cross-cortex activity of the brain [247,253,254].

In the context of surgical training, DUST topology has been used for EIL haptics-enabled training scheme. For this, the expert and the novice surgeons control the two leader robots while the follower robot is to conduct tasks on a physical phantom of the surgical site designed for training. Using this architecture while the expert surgeon is conducting the task, using a rigid virtual fixture implemented at the novice's side, the novice would learn how to manipulate and control. Besides, when the novice surgeon is performing the task, the expert surgeon can provide corrections and cues since there is a haptics coupling between the two leader robots. In addition, when the two surgeons conduct the task, the system can adaptively tune the authority level and the stiffness of the virtual fixture. In this way, if the novice surgeon is generating high-quality motions (which can be quantified through comparison with the expert surgeon's motion on the fly and during the operation, and based on measures such as trajectory differences, smoothness, etc.), the system can allow for higher authority and lower stiffness of the dynamic virtual fixture. Examples of such designs can be found in the literature [4,163,258,260–268]. In [258], for instance, an expertise-based surgical training framework was introduced, in which novice trainees will be guided with haptic guidance, and when the acquiring higher expertise, the force feedback would gradually switch from the haptic guidance toward environmental force reflection. Fuzzy logic is used in this work to fuse several objective measures about the expertise of the novice surgeon.

Dual-User Redundancy Control (DURC)

The difference between this topology and DUST topology is that here the two operators control different degrees of freedom, joint, or motions of the follower robot, while in DUST topology the two leader motions are fused to control the Cartesian motion of the follower robot in the task space. This feature improves the operability of the follower robot for complex environments and enables multitasking due to the separate assignments of the degrees of freedom of the follower robot to the operators [279]. DURC topology is used for tasks such as robotic telerehabilitation [280] in which a 6-DoF arm motion of the patients is to be controlled using two 3-DoF leader systems controlled by two operators to perform complex rehabilitative tasks. It is crucial for the leader robots to operate in synchrony to coordinate arm movements safely (for which a local intelligence can be of high benefit). Other examples can be found in [281,282,310].

Section Vision

Having established various topologies of teleoperation systems, their design, usages, challenges, the future direction of this field of research should be evaluated holistically. We have shown that several topologies of telerobotic architecture have allowed for a large spectrum of tasks and applications, which would not be possible in the absence of such technology. Among all the challenges facing teleoperation, uncertainty, and time delay are still the most prominent problems resulting in asynchrony, instability, and task failure. New advanced intelligent control architecture, predictive models, and the use of autonomous agents can help to mitigate the mentioned challenges.

## 3. Autonomy Levels

This section aims at describing different levels of autonomy and embedded intelligence in various telerobotic solutions in the context of teletherapy. We consider a gradual increase from direct telerobots with no intelligence to fully-autonomous telerobots.

The gradual increase in autonomy was first defined by Sheridan et al. in 1978 [311] with ten levels of autonomy (including none and full autonomy). Conway et al. reduced this scale in 1990 [312] to (1) direct continuous teleoperator control, (2) shared continuous teleoperator control, (3) discrete command control by the human operator, and (4) supervisory control. In 2016, Nichols et al. [313] modernized the terms into bilateral teleoperation, shared control, supervised control and supervisory control, modified the order of shared and supervised control, and introduced a new level, i.e., traded control, which corresponds to a mix in time between teleoperation and shared control, and formalized a generic software architecture related to that. Abbink et al. [314] and Yang et al. [304] also revised these terms (with six levels). In this paper, we propose to bring some refinements to the scale proposed in [313], taking into account [304,314], and proposing a rising score of autonomy order for telerobotic systems (details can be found below and in Figure 7):

1.  Bilateral teleoperation: features any exchange of position (and force) between the leader and the follower robots (only position exchanges were envisaged in [313]) and uses a SL/SF topology defined in Section 2.1;

2.  Shared control which can be split in two subcategories:

    *   Assisted Shared Control: the operator is assisted with an auxiliary feedback added to the teleoperation such as continuous vibratory feedback, or virtual fixture. This is in agreement with the definition of "shared control" in [312,313];

    *   Multi-user Shared Control: corresponds to architectures which include several operators sharing the control of the same telerobot;

3.  Traded Control where the users alternate between bilateral teleoperation and shared control [313] or supervisory control [314]. They may switch assisted/automated control off when it does not comply with the task requirements.

4.  Supervisory Control, where some high-level information (parameters and/or offline programming) is sent to the follower robot to be reproduced with some degree of controlled autonomy, knowing that in main medical applications, there are no very long-distance constraints.

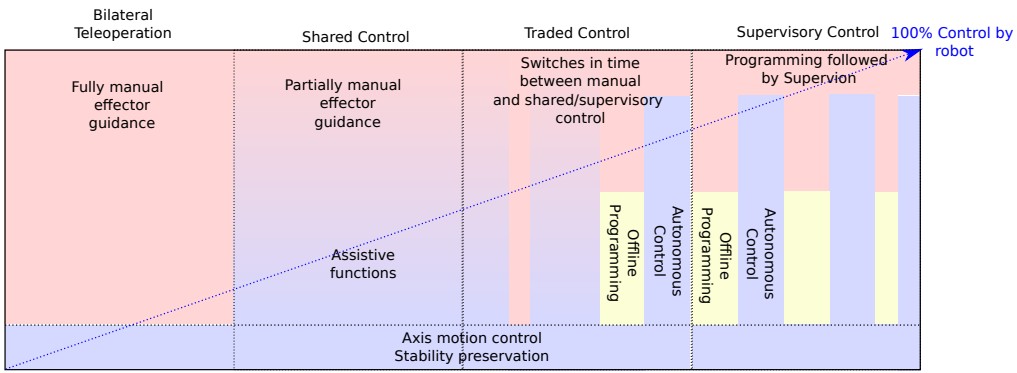

**Figure 7.** Autonomy levels, from teleoperation (**left**) to supervisory control (**right**) in teletherapy applications.

### 3.1. Bilateral Teleoperation

As mentioned in Section 2, in teleoperation architectures, the follower robot mimics the actions of a leader robot controlled by an operator. In surgical applications, typically, two arms are independently controlled by both operators. Thus the system is indeed two independent SM–SS architectures (see Section 2.1). A complementary human–robot interface (HRI), often featuring pedals, enables the operator to move the third arm handling the endoscope.

Still recently, many generations of surgical robots have only relied on position-position exchanges: the follower tracks the position of the leader and reciprocally. For instance the

da Vinci robot does not enable force feedback (a solution to circumvent this limitation on this specific robot has been recently proposed in [114]). Yet, as detailed in Section 2.1.1, some reports have been regularly published about the usefulness of haptic feedback in specific medical applications. Though teleoperation systems have featured force feedback for a long time in other applications (spacial, submarine, mobiles robots, ...) [4,54,206,312]. This lateness in the medical field can be explained, on the one hand, by the patents of intuitive surgical that blocked the development of this market, and on the other hand, by the difficulty to measure the interaction forces during surgical tool-patient interactions, while remaining compliant with medical sanitation laws, and to efficiently transmit it to the surgeon. Hopefully, force feedback has been progressively integrated into the next generations of medical robots.The very first haptics-enabled commercialized surgical robot is considered to be the NeuroArm (IMRIS, Inc., Minnetonka, MN, USA): a telerobot designed for MR-guided biopsy and stereotaxy embedding two 3-DOF optical force sensors. It is controlled remotely by a surgeon from a robot workstation featuring several display monitors and two haptic devices. It was first introduced as a project in [315], used for a first neurosurgical procedure in 2008. In 2009, the MiroSurge telerobot from the German Aerospace Center (DLR) had similar functions to ZEUS and LARS, but with force feedback [49,50]. Since, other robots with force feedback were introduced, such as the AQrate® System (KB Medical) for minimally invasive spinal surgery, Avatera (Avateramedical GmbH, Jena, Germany), MiroSurge, Revo-I®, Senhance (2017, TransEnterix, Morrisville, NC, USA), the CMR Surgical (Cambridge, UK) Versius, the REVO-I RAS system (Revo Surgical Solutions, Seoul, Korea), or the Stryker's Mako platform for total knee and hip replacement [52]. Recent works published promising solutions for medical applications, providing haptic feedback with forces about 4 N with a resolution of 0.03 N and stiffness about 3.6 N/mm with a resolution of 0.025 N/mm [316].

One of the great features of bilateral teleoperation is the opportunity to provide scaled manipulation [317], to manipulate objects at a smaller scale and to magnify rendered forces. In medical applications, the motion of the follower effector is often scaled down to perform delicate and precise tasks. Yet, Cassilly et al. showed that motion scaling reduces the number of errors at higher magnifications, but could also increase the task completion time [318]. Indeed, the drawback is that the attainable space is proportionally reduced, which prevents the surgeon to perform larger movements to navigate between targets. This is why a clutching approach has been rapidly integrated into medical robots: when clutched, the follower does not move and the surgeon can move the leader back to extend his movement once unclutched. However, this is repetitive and time-consuming. Recently, Zhang et al. introduced a self-adaptive motion scaling mechanism that adapts to the user skills and the task requirements [319]. We cite this work here for readability concern, but it corresponds to shared control solutions detailed in the next section.

Bilateral teleoperation is also applied for remote ultrasonography [320,321]. Nowadays, over a quarter of emergency admissions requires an ultrasound examination for preliminary diagnosis purposes. This is a low-cost radiation-free examination technique, which implies that the physicians remain very close to their patients to position the ultrasound probe on the targeted anatomic area. Since the late 1990s, several solutions of telerobotized ultrasonography have been developed to compensate for the lack of ultrasound experts in medically isolated settings (see, for instance [18]). Two concepts have been proposed to teleoperate the ultrasound probe: either with a robotic arm or with a light-weight specific robot maintained on the patient by a paramedic, while the physician remotely actuates the probe to collect and then later analyze the ultrasound images. Ultrasound probe guiding robots can also be used for enhanced ultrasound examination as an assistance [16,17], such as described in the following shared control section.

Nowadays, bilateral-control based solutions are mainly studied in terms of stability [64] (or passivity [182,322]) and transparency [182,323] taking into account communication delays [182,274,320], packet drops, disconnections, data quantization, actuator

saturation, operator dynamics [168] to provide safer low-level architectures that can then be used with embedded operator assistance intelligence, to generate the following categories.

### 3.2. Shared Control

The shared control paradigm has been proposed in various forms to assist teletherapy operations. It mainly helps improve the operators' sensorimotor and spatial reasoning skills. Using this strategy, the operators control the remote robot, but they are assisted in performing the task. This approach is also referred to as cooperative or collaborative teleoperation, as it can be seen as a collaboration between either several operators (such as in the aforementioned dual-user topologies) or between a "robot intelligence" and the operator. In the second case, depending on the amount of assistance provided to the human operator, the cooperation is performed at different levels, ranging from low-level sub-tasks (kinematic transformations, environment motion compensation [324], force filtering, motion planning [325]) to high-level tasks (situation analysis, planning generation, and decision/proposal making [313]). There is no single definition for shared control [314]. We propose to divide the shared-control-based solutions into two types of strategies: those based on task decomposition and those based on authority blending.

### 3.2.1. Task Decomposition Based Shared Control

The first category of shared control strategies, shown in Figure 8, lies on the task-decomposition approach, which consists of decomposing the task into several sub-tasks, where some of them are performed by the user, and others are automated for assistance purpose [16,313,325–327]. This decomposition can be performed by automating the most difficult part of the motions while leaving the operators free for safe motions and agency. A motivation for reducing their maneuverability is the decrease in their cognitive load. This approach is also called dimension reduction in [313], where the palpation task was decomposed into three chained sub-tasks: the robotic agent decides for the palpation locations, then the operator controls the palpation during the downward stroke until the force threshold is reached, and at which point, the robotic agent controls the upward palpation stroke . Thus, the operator is solely responsible for imparting forces. Another application is proposed in [327] where a flexible manipulator creates dynamic trajectory plans automatically. The surgeons only need to define the dissection trajectory with a few markers, while the system generates a trajectory based on the feedback from the stereo endoscope system, with consideration of the deformation of the tissues.

Motion compensation is another application of task decomposition. For instance, the steady hand cooperative control was introduced at first in 1999 in [66]: a force sensor detects forces exerted by the surgeon on the leader handle, and the follower robot moves to provide smooth, tremor-free precise positional control and scaled force feedback. This medical application would be extremely difficult and unsafe (if even possible) for the patients without this assistance. For instance, in [328], the beating heart motion compensation is decomposed into three tasks; (1) motion synchronization, (2) image stabilization, and (3) shared control. In [211], for lung motion compensation during needle insertion, an impedance-based control was used. This approach has been enhanced in 2017 in [16] for tele-echography purposes where an impedance-controlled teleoperation system compensates for the natural motion of organs such as heart, chest movements uniquely via appropriate parameter adjustment in the desired impedance models, without requiring any direct measurement and/or online prediction of the organ's motions, and even in the presence of communication delays and modeling uncertainties. An enhanced version has been proposed in [329] and a multi-user version is available in [324] for training or cooperation purposes.

In the particular case of multi-user training, the number of sub-tasks is increased proportionally. In this case, in addition to the aforementioned tasks, a new task adds position guidance to the trainees during the training procedure, and another one provides force feedback to the additional operators regardless of their levels of authority over the follower robot.

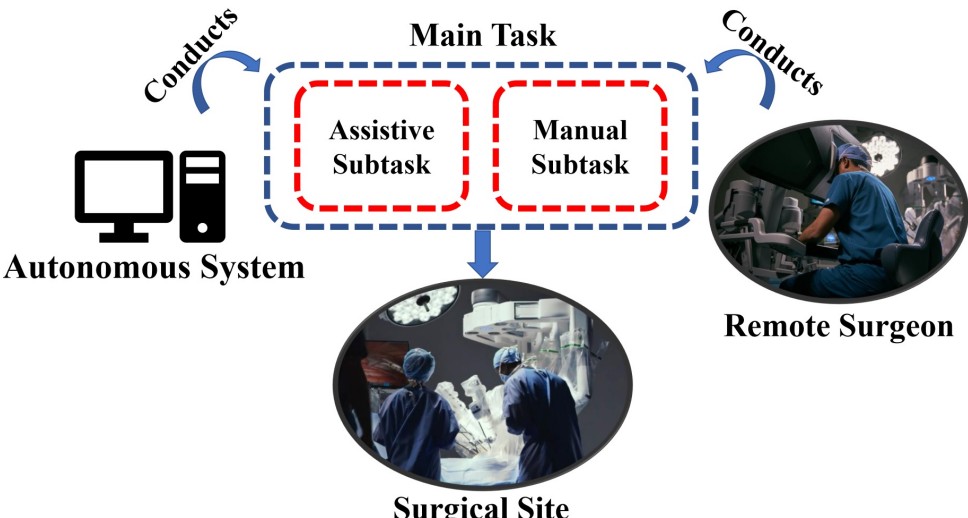

**Figure 8.** Shared control based on task decomposition, main principle (Images ©2020 Intuitive Surgical, Inc.).

Virtual fixture functions described in Section 2.1.2 provide guidance or inhibition feedback that is generated in a subtask synchronized with the operator's manual subtask. These features add safety and directional operations through software. For instance, in [326], a subtask is responsible for the online generation of some VFs using interaction force measurements, preoperative information (such as CT scans), and intraoperative information (such as body motion). A 2014 detailed survey about VFs is available in [67].

Traditionally camera control is conducted manually by the surgeon during the surgical workflow. An autonomous robotic system can help to decompose the controlling of tools and camera separately with minimum interruption of the continuous operation of surgical flow. In the past years, several shared control frameworks have been proposed by semi-autonomous systems for camera control. A review of the state of the art of autonomous camera control systems in surgical applications is conducted in [330], in which various techniques are introduced for autonomous control of the camera based on eye-gaze tracking, instrument position in space, and image-Based scene tracking. For example, eye gaze tracking is used to automatically center the viewpoint of a laparoscopic camera regarding the user's point of gaze in [331]. As another relevant topic, path planning methods for the camera control have also been evaluated in the literature, such as an algorithm based on rapidly exploring random tree (RRT) [332]. The development of autonomous camera control of the da Vinci Surgical System is investigated in [333]. It can generate autonomously-centered and zoomed viewpoints by keeping the surgical tools in the camera's field of view. The aforementioned technique has shown reduced user-perceived workload and increased efficiency, and progress. Besides, it should be noted that autonomous camera control can be very beneficial for the initial stages of surgical training, when the trainee is becoming familiar with robotic surgery, and thus simplifying the control can be significantly beneficial for the early stages of training. In this regard, the autonomous camera navigation during the robotic surgical training is investigated in [334] where the experimental evaluations suggested improved performances and efficiency of training with robot-assisted surgery. In summary, it can be mentioned that autonomous control of the camera can be considered as one modality of shared control (between the machine and the operator) to reduce the cognitive workload, enhancing ease-of-use and providing more safety and consistency of surgery.

Task decomposition for a human–machine shared control strategy is also used in various medical applications such as the control of a wheelchair [325]. In this case, a brain–machine interface using steady-state visual evoked potentials is introduced to guide the wheelchair while a vision-based algorithm provides simultaneous localization and mapping (SLAM) to help with navigation among the obstacles. Another relevant application is

myoelectric control of prostheses [335,336], which allows users to recover lost functionality by controlling a prosthetic robotic device with their remaining muscle activity. In [337], computer vision (for autonomous object recognition) and mechanomyography (to estimate the intended muscle activation) data are fused to conduct a shared control that predicts user intent for grasp and then realizes it. In [338], once the user establishes a pre-contact between the robotic hand and an object (manual task), the shared controller optimizes the actuation of the fingers of a robotic hand to maximize the contact between the hand and object to obtain full-contact (robotic task).

In 2018, Watanabe et al. proposed a shared-control based solution to perform semi-autonomous suturing with two robotic arms. In this procedure, the operator inserts the needle into an organ using one of the follower robots when follower A1 was directly controlled through a bilateral teleoperation architecture and follower A2 automatically grasps the tip of the needle and pulls it out from the organ, and automatically hands the needle back to follower A1. The operator repeats the same for each throw. An estimation of the force involved in the interaction of A1 with the organ is used to trigger the start of the A2 sub-task. This way, the completion time was decreased by an average of 20% in total. This is an interesting mixture of autonomy levels. Ref. [339] highlights that finding the point of puncture or holding the needle are activities that require some adaptation or manual correction by the surgeon, with risks of incorrect suturing and subsequently postoperative complications (calling for more advanced research related to autonomy).

### 3.2.2. Shared Authority Blending-Based Control

In this second category, the authority of control of the follower robot is distributed among the (real human or virtual) operators, with a balance that can vary in time. They share a complementary part of the control authority that is manually set on-line by one of the users or dynamically managed. Such authority blending is applied on multilateral teleoperation topologies (ML/SF and ML/MF, see Sections 2.3.1 and 2.3.3), out of them, dual-user topologies are the most studied ones (see Figure 9). Typical applications of dual-user haptic systems are for cooperation or training purposes, where both humans control the system through the shared control structure and are both provided with haptic feedback. It is also introduced as shared autonomy when a human and an autonomous system work together to achieve shared goals [340,341]. We will distinguish linear and nonlinear blending approaches in this section in dual-user contexts and then in multilateral ones (more than two users).

Considering the training application, the interest of haptics-enabled computer-based training systems for gesture training is that the follower is virtual (a simulation of the tools and the environment and their interactions) such that the trainees can rehearse as many times as necessary on the same exercise while being provided with objective assessment. Even if the aid from an experienced person can accelerate the training, this person can only guide the trainees "from the outside of the simulation", through a hand-over-hand guiding. As for traditional hands-on training, this approach does not permit both users to feel and dose the forces to apply to the tools as they share the same interaction. Dual-user training systems permit the trainers to get into the simulation (or even the manipulation of a real surgical robot), with force feedback for trainers and trainees.

### Linear Blending

Shared control based on linear blending has been first introduced by Nudehi et al. , in 2005 in [268] for MIS telesurgical training, to allow for hands-on training of novice surgeons based on the skills of experts. It consists of a variable $\alpha \in \mathbb{R}$ with $\alpha \in [0,1]$, such that the follower robot velocity, $v_s$, is a weighted sum of both user input velocity $v_{m_i}$ with $i \in \{1,2\}$ (*m* and *s* corresponding respectively to master (leader) and slave (follower)) such that:

$$v_s = \alpha v_{m_1} + (1-\alpha)v_{m_2} \tag{1}$$

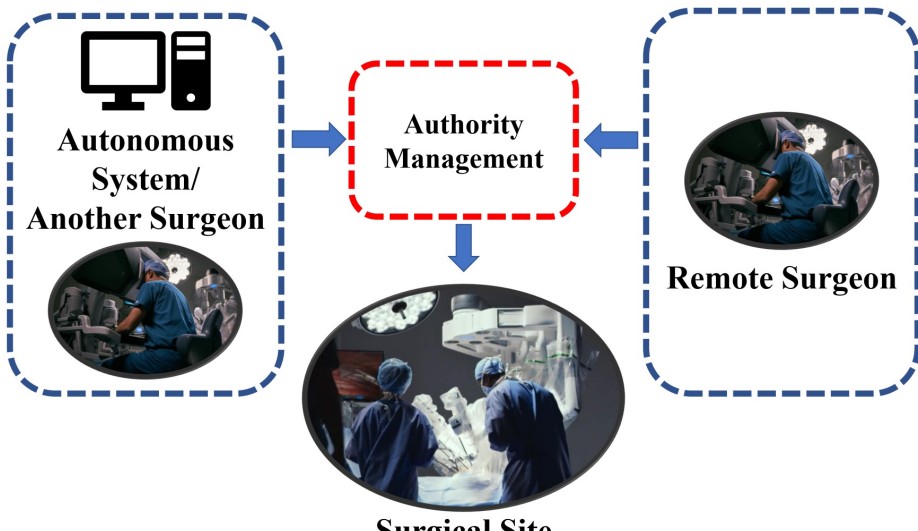

**Figure 9.** Shared control based on authority management, in the specific case of a dual-user topology (Images ©2020 Intuitive Surgical, Inc.).

Thus, when $\alpha = 1$ (resp. 0), user #1 (resp. #2) teleoperates the follower robot and user #2 (resp. #1) inputs do not affect the follower robot motion. When $\alpha \in [0,1]$, the follower velocity reference is mixed according to Equation (1) so that both users share the leading of the follower according to the level of $\alpha$. When $\alpha = 0.5$, both users' actions are equally balanced.

There is no unified approach in the literature on the haptic feedback blending. In [268], the only feedback received by the users from the tool-environment interaction was visual, employing a remote camera image displayed on a local monitor. In practice, they fell in their hands a force proportional to the difference of position difference between them:

$$
\begin{aligned}
f_{m_1} &= K(1-\alpha)(x_{m_2} - x_{m_1}) \\
f_{m_2} &= K\alpha(x_{m_2} - x_{m_1})
\end{aligned}
\tag{2}
$$

where $x_{m_i}$ and $f_{m_i}$ are respectively the position and the force feedback of master (leader robot) $i$, with $i \in \{0,1\}$, $K \in \mathbb{R}^+$.

This provided the "following" user with some virtual force, guiding him/her toward the "leading" one, but it did not help novice users feel and learn the tool-environment interaction forces $f_e$. Therefore, this kind of training system can only be used to train users on motions, not on efforts. The same limitation applies to [263] where the tool is virtual. Khademian and Hashtrudi-Zaad overcame this limitation in [266] with two architectures: the complementary linear combination (CLC) and the masters correspondence with environment transfer (MCET). In both architectures, master (leader) controllers are fed with reference signals that are linear combinations of desired velocities and feedback forces. For the CLC architecture, they are:

$$
\begin{aligned}
v_{m_{1_d}} &= \alpha v_s + (1-\alpha)v_{m_2}, \quad f_{m_{1_d}} = \alpha f_e + (1-\alpha)f_{m_2} \\
v_{m_{2_d}} &= (1-\alpha)v_s + \alpha v_{m_1}, \quad f_{m_{2_d}} = (1-\alpha)f_e + \alpha f_{m_1}
\end{aligned}
\tag{3}
$$

where $v_{m_i}$ is the velocity of master $i \in \{0,1\}$, $f_{m_i}$ the force applied by the user on master $i$, $v_{m_{i_d}}$ is the desired velocity for master $i \in \{0,1\}$, $f_{m_{i_d}}$ the desired feedback force for master $i$, $f_e$ the interaction force between the environment and the follower robot. For the MCET architecture, the desired feedback force is half of the environmental force for both users:

$$v_{m_{1_d}} = v_{m_2}, \quad f_{m_{1_d}} = \frac{1}{2}f_e$$
$$v_{m_{2_d}} = v_{m_1}, \quad f_{m_{2_d}} = \frac{1}{2}f_e$$

$$(4)$$

Unfortunately, as visible in previous equations, both architectures feed users with a distorted tool-environment force. As a matter of fact, in numerous works, confusion for the user between the interaction forces (reflected from the follower robot interaction with the remote environment) and those generated for guidance and training is possible due to their mixing. In practice, in the low-level control, a force (or a torque) is generated by an actuator, and this force is generated so that the haptic device follows a combination of desired velocities and feedback force. It is then difficult for the user to differentiate the force generated to control the velocity and the feedback force during transient periods. During free motions, the force feedback typically corresponds to some guidance if any. During immobile interactions, as there every device is steady, the force feedback should reflect $f_e$ (no follower interaction), Trouble in sensing correct force feedback may arise when switching between free motion and immobile interaction phases and during mobile interactions (while touching soft materials for instance). In this last case, feeling the right environment force requires the leader device to be moving at the exact desired velocity. As advised in [342], visual help should be provided to the "following" users to correctly position their device as the "leading" ones, during tool-environment contact phases to mitigate this challenge.

Another linear-blending architecture is defined by [208], using an intrinsically passive architecture that enables both users to experience a full $f_e$ force feedback. One can find other recent uses of linear blending for dual-users systems in [343,344]. In practice, the authority blending should be ensured taking into account variable communication delays, packet loss, data duplication, and packet swapping [308], and actuator saturations [295].

The linear blending approach also increases efficiency; Saeidi et al. demonstrated in [345] that a 2D-pattern cutting task with partial blood occlusion could result in a work time diminution compared to a fully manual task. This experiment highlighted the interest of a confidence-based shared control strategy with an adaptive blending. The aforementioned works on brain-interface for wheelchair navigation also make use of adaptive linear blending [325], which permits to dose the trajectory between the desired direction and the avoidance of obstacles. However, it remains important to leave some control on the blending to the users. Indeed, Gopinath et al. introduced in [340] a theoretical framework based on optimal control to optimize in real-time the linear blending according to users' preferences (i.e., own cost functions). It resulted that the amount of assistance was voluntarily lowered as some subjects favored retaining more control during the execution over better task performance.

Nonlinear Blending

Nonlinear blending has been introduced by Ghorbanian et al. by splitting the dominance into two factors $\alpha$ and $\beta$ [308]. Here, $\alpha$ balances the authority between both users while $\beta$ sets their supremacy over the follower robot. This provides an additional degree of freedom in the authority mechanism that is leveraged employing a nonlinear relation between $\alpha$ and $\beta$. Finally, only one parameter has to be set, which provides finer control of the authority. In the nonlinear blending category, we distinguish linear-blending based strategies where $\alpha$ is determined using nonlinear mechanisms (we will refer to these solutions as "nonlinear over linear" solutions (NL/L), and full nonlinear solutions.

In the NL/L category, Shahbazi, et al. [206] propose an adaptive fuzzy-logic-based design to dynamically set $\alpha$, according to the acquired expertise of the novice surgeon. It is evaluated on the fly from various measures by comparing the performance of the novice user with that of the expert. In [208,346], the adaptive authority adjustment (AAA) function

is proposed to automatically revert, in trainee evaluation mode, the authority from the trainee to the trainer in case of undesired behaviors of the trainee (as soon as the trainer deviates from the trainee's trajectory to rectify it). Following the same need of attributing the authority to the right user, the solution introduced in [347] reverts the authority to the trainer when he/she applies a force greater than a predefined one on his/her haptic device. In [348], in the context of surgical grasping, the robotic controller and the human inputs are linearly blended with a balance based on the computed confidence in the identification of the grasped object by an online identification of the grasped tissue. This strategy outperformed the regulation of the grasping forces to the desired target force compared to manual control.

We cite here some authority blending solutions that are not based on the Equation (1) linear approach. In [247], adaptive blending is applied for robotics-assisted mirror rehabilitation therapy with the therapist-in-the-loop (TIL) approach, realizing a nonlinear Assist-as-Needed Therapy (ANT), see Section 3. A haptic-negotiation model that dynamically mixes the velocity and haptic feedback exchange between a wheelchair driver, an assistant, and the wheelchair follower controller is proposed in [349]. At first, a real human assistant trains a Gaussian process (GP) regression model that will then act as a virtual assistant.

Authority Blending with More than Two Users

Authority blending has been extended to multilateral (ML/SF) topologies for generic purposes in [218], for the integration of a human assistant (forming so a trilateral topology, in [247,349]), for cooperation purpose in beating-heart surgery (in the aforementioned [324]), and for training purposes with one (dual-user [218]) or several trainees (multi-user [342]) with real tool-environment force feedback simultaneously felt by every user. Multi-trainee architectures allow several trainees, along with a trainer (potentially several ones) to use and learn on the same haptic training simulator (or real follower device) at the same time. This avoids the trainer to repeat the same gestures for each trainee. One trainee can perform a demonstration for evaluation purposes simultaneously towards the trainer and the other trainees, who can observe it, which can be interesting from a didactic point of view. In [218], a linear blending is proposed so that operators experience force feedback from the follower-tool-environment interaction force corresponding to their authority level. An impedance-based control methodology is adopted to guarantee the passivity and so the stability of the system in presence of communication delay. In [342], a trainer can also demonstrate a particular motion trajectory (and trainees follow this demonstration with haptic feedback). As another sample application, the works exposed in [324] permit shared collaboration and training between $n$ operators with non-oscillatory feedback in a beating-heart surgery context. In this work, multi-user linear blending is proposed with a second force scaling factor $\beta$ that permits to enable position guidance for trainees in demonstration mode (called fundamental training) independently from the authority attribution parameter $\alpha$. This guidance is provided as a virtual force generated by a virtual fixture designed to guide the trainees along the right path of the surgery.

3.2.3. Shared Control Synthesis

Table 1 synthesizes a wide range of works in the literature. This table does not include all existing works in the literature due to the significant size of existing research. However, it provides examples of each category. The table focuses on medical application, the level of the task performed by the assistance, the activity of the assistance, the presence of haptic feedback, the corresponding topology, the autonomy category, and the solutions for shared control. This table shows that solutions are provided in the literature from low to high levels of control layers, for a wide variety of applications.



**Table 1.** Shared control samples for teletherapy applications.

| Papers | Application | Task Level | Assistance | Haptic Feedback | Topology | Category | Solution |
|---|---|---|---|---|---|---|---|
| [313] | Surgical Manipulation (palpation) | High-level | Decision Making | ✔ | | | Dimension Reduction |
| [327] | Surgical Manipulation (dissection) | | | ✘ | | | |
| [326] | | | | ✔ | | | Virtual Fixtures |
| [328] | Surgical Manipulation (tracking) | | | ✘ | SL/SF | Task Decomposition | |
| [329,350] | Surgical Manipulation (tissue contact) | Low-level | Motion Control | | | | Motion Compensation |
| [16] | Tele-echography | | | ✔ | | | |
| [348] | Surgical Grasping | | | | | Authority Blending | Linear Blending |
| [345] | Surgical Cutting | | | | | | |
| [325,340] | | High-level | Decision Making | ✘ | | | |
| [247] | Assistive Rehabilitation | | | | | Task Decomposition | Virtual Fixtures |
| [349] | | | | | Trilateral | | Nonlinear Blending (virtual spring-damper) |
| [206,208,268,269,308, 343,344,347,351] | | Low-level | Motion Control | ✔ | | Authority Blending | Linear Blending |
| [352] | Surgical Training | | | | Dual-user | | Nonlinear Blending (cubic polynomials) |
| [218,324] | | | | | ML/SF | | Linear Blending |

Besides medical applications, there exist several other works that relate to shared control strategies for other applications. A few of them are cited below, selected for their interest. For instance, in [353], authors have developed virtual fixtures to help operators grasp objects in a scene, resulting in an improvement of 20.8%, 20.1%, 32.5% in terms of completion time, linear trajectory, and perceived effectiveness, respectively, between the proposed approach and standard teleoperation. As surgical grasping is a common task, such improvement should help enhance teletherapy applications.At a higher control level, Javdani et al., highlighted that the assistance provided by the autonomous system requires "knowing" the user's goal to be effective. Solutions found in the literature based on a predict-then-act model (see [341] for corresponding references) may not be effective due to the intrinsic limitations of prior prediction of human intentions. The authors then proposed a real-time observation based on a Markov decision process with online optimization to determine the most probable current goal of the user. Experiments showed that this approach *"reduced the task time compared to predict-then-act, required less user input, decreased user idling time, and resulted in fewer user-robot collisions"*. Knowing the high cognitive requirement of surgical operations, such an approach should also be interesting in a teletherapy context. In [354], the assistance is achieved through programming by demonstration (PbD). The assistance and user inputs are blended according to two confidence indices (on user and assistance) computed online that determine their relative weight. The assistive system uses a Gaussian mixture model (GMM) to represent the task, and the desired state associated with a confidence level is obtained using a Gaussian mixture regression (GMR). Experimental results on teleoperated object manipulations showed a light preference for the trained solution. Authors conclude that this is a promising approach that still requires investigation to become more effective. Other similar shared autonomy formulations can be found in [355,356]. We can also cite the works of Zakerimanesh et al., that permit multilateral teleoperation (using nonlinear authority blending) for remote applications featuring time-varying communication delays, actuator saturations, nonlinearity in the dynamics (which corresponds to a common teleoperation practical situation), and more particularly for follower robots with redundancy, which is interesting with medical robots as this redundancy is typically used to avoid collisions with the staff and other devices around the patient. Shared control can also be used for some severely disabled people with brain–computer interfaces (BCI). Non-invasive motor imagery-based (MI-based) BCI, relying on Computer Vision and electroencephalography (EEG) has been proposed in [357] to control a robotic arm. Experiments with five subjects were performed. Users only needed two different mental tasks to reach the surrounding area of the target. The grasp of the target was then realized employing a depth camera. The success rate was above 70% with no specific user training.

A 2017 survey about shared control, not limited to teletherapy applications, is provided in [358] with more details and highlights potential great enhancements in shared autonomy medical applications in the future.

### 3.3. Traded Control

In traded control (introduced by Matijevic et al., in [359] and Hayati et al., in [302], the follower robot sometimes operates autonomously (performing for instance a task of bone drilling [360] or surgical knot tying [361]) and, sometimes, teleoperatively to facilitate the trade-off between teleoperated and autonomous modes. In practice, it can be considered as an intermediary remote control strategy between supervisory or shared control/bilateral control.The surgeon can use automation features when they are efficient, and can take back the full control of the follower robot if necessary. In 2018, Watanabe et al., highlighted that "automating ubiquitous surgical subtasks such as suturing makes surgery more efficient, e.g., reduction of surgeon fatigue and/or surgery time", but "conducting fully-autonomous surgical task remains risky due to individual difference in human body" [362]. What differentiates this category from shared control is that the level of autonomy is discretely switched in time on demand of the surgeon. In shared control,

the autonomy level is continuously moderated, manually, or automatically. As declared in [313], this collaboration strategy is not commonly used in the literature. A search in July 2020on "$x$ control" AND teleoperation with $x \in \{$bilateral, shared, traded, supervisory$\}$ in Scopus (See https://www.scopus.com) database provided respectively 432, 198, 14, and 104 results, respectively. Moreover, Scopus outputs only one paper corresponding to traded control in a medical context: [313]. The same requests on the IEEE database (See https://ieeexplore.ieee.org) provided only seven results for traded control and only two in the medical context, from the same authors as [313]. This can show great potential for investigation on the context of traded control.

It should be noted that Parasuraman et al. [363] introduced a four-stage model of human information processing in teleoperation tasks (i.e., (1) information acquisition, (2) information analysis, (3) decision and action selection, and (4) action implementation) where the functions in each stage can be automated. For instance, automation at stage 1 can be performed by providing active sensors directing themselves towards the area of interest or by providing adaptive noise filtering. In stage 2, one can envisage predictive functions to bring up more useful information. In stage 3, an expert system may help the surgeon choose the best strategy, and in stage 4, the computer can relieve the surgeon by realizing itself a repetitive or dangerous task (automatic palpation for instance). This model is aimed at guiding roboticists in their design of teleoperation architectures. The multilateral manipulation software framework (MMSF) was introduced in [313], aimed at structuring collaborative tasks to facilitate rapid development of human–robot collaboration models. This framework is to be used upon low-level robotic software frameworks such as ROS. It has been applied for the segmentation of a relatively stiff tissue (e.g., a tumor) from the surrounding soft tissue task.

Traded control is also useful for telerehabilitation purposes. In this regard, the concept of learning from demonstration is utilized for robotic rehabilitation [364–367]. This strategy encompasses two distinct phases: during the first (therapist-in-loop) phase, the therapist interacts directly with a patient through bilateral teleoperation. During the second (therapist-out-of-loop) stage, the follower robot displays the learned therapeutic behavior to the patient (for example via an impedance control loop).

### 3.4. Supervisory Control

Supervisory control is when the operator asks the follower robot to perform autonomous high-level tasks under the supervision of the operator who can interrupt them. Functions provided by supervisory control include planning, teaching, monitoring, repairing, and learning from experience [368]. This approach derives from contexts with very high transmission delays and low bandwidth, such as in space and submarine applications [369], which could result in the inability of the operator to provide the appropriate commands on the fly (due to the affected causality) and could result in instability and divergence. It has been reused repetitively for swarm robotics where the followers are numerous so not directly controllable by the operator [370].

Supervisory control is now used to get performance on "routine tasks" [368], such as autonomous parking functions in autonomous cars. In the surgical domain, this is a very active topic of research and the corresponding research is accelerating. There are several technical challenges to be addressed including tissue deformations and mobility, unpredictable scene changes resulting from cutting, suturing, or cauterizing operations. There are some legal structure which should be also implemented for autonomous surgery before its wide deployment [371].

In 2016, Shademan et al., demonstrated [372] that supervised control performed on the smart tissue autonomous robot (STAR) was *"not only feasible but also, by some metrics, surpassed the performance of accepted surgical techniques, including robot-assisted surgeries (RAS), laparoscopy, and manual surgery"*. In 2020, Liu et al. report the use of supervisory control in orthopedic operations with a RoboDoc (THINK) robot [373]. Operations are performed in two stages: surgical planning by the surgeon considering pre- or intra-

operative patient information from CT–MRI scans or 2D/3D fluoroscopy, and then robotic autonomous execution. More examples of surgical autonomy can be found in [372,374–376] and references therein.

### 3.5. Discussion About Robotic Autonomy

Robots have been progressively moving the frontier of tasks that can be conducted remotely, collaboratively, or autonomously. The raising autonomy of robots helped humans secure and optimize basic tasks but also opened the way to new applications which are hardly-feasible or not feasible manually such as in several micro-scale surgical tasks [51,53,377,378], some maxillofacial surgeries [360], some dental surgeries [93], and some ophthalmic operations [148,378]. There remain several operations that are challenging for robots especially those including the unpredictability of the environment (such as autonomy on soft and deformable tissues) and those requiring a high level of domain knowledge for fast reflexive reactions (in extremely sensitive operations). Research in robotics has augmented sensorimotor skills of human and at the same time provided insight on ways to cooperatively perform tasks which are not feasible by robot only or human only.

Thanks to the arrival of new telecommunication paradigms such as 5G and beyond, a new generation of teleoperated applications can be envisioned with ultimate fidelity and safety of force feedback thanks to unprecedented quality of service (very low bandwidth, delays, and jitters) of novel communication systems and performance of advanced control algorithms.

It can be imagined that the level of autonomy of medical telerobots and patient safety will progressively increase along with technological progress.

Based on the current trend, we can envision that autonomous telerobots will not aim to replace surgeons but to expand human capability through more efficient, adaptive, high-level, and safe operation offered by robotic dexterity and artificial intelligence.

### 4. Conclusions and Future Directions

Telerobotic technologies have opened new doors to investigate the future of the modern healthcare system when smart and connected infrastructures can address several existing challenges. Telerobotic systems were initially designed to extend human access to remote environments and bypass physical barriers, such as deep water, space, and radiations. With the utilization of modern medical telerobotics, other potentials of this technology have been realized. With the use of this technology, currently, many clinical tasks are possible, which were not realizable before the use of telerobots.

In this paper, we discussed several applications of telerobotic technologies, including telerobotic surgery and telerobotic rehabilitation. We have identified sensory augmentation and motor augmentation benefits of intelligent telerobotic systems; when using a sensorized tool, the awareness of the operator (e.g., a surgeon) about a particular task has reached a level beyond human competence, and the resolution of movement control has been significantly improved. These benefits are mainly discussed in the context of telerobotic surgery, for which a successful commercialized example is the da Vinci surgical system. We have also highlighted other medical applications of telerobotic systems, in particular, telerobotic rehabilitation for delivering physical therapy and assessment of patients with motor disabilities at remote locations when patients are at home and clinicians are located at far distances in clinics. We have explained how such technologies have attracted a great deal of interest, especially after the COVID-19 pandemic, due to their power in augmenting the current telemedicine, which relies only on visual and verbal interaction between the clinician and the patient. This paper also discusses the existing challenges about the use of telerobotic systems, including the lack of force feedback in several medical telerobotic technologies due to the challenges associated with instrumentation and stabilization. We have discussed that with the use of advanced instrumentation technologies and intelligent

stabilizers, the next generation of medical telerobotic systems can be enabled with force feedback modality.

In addition to the above, this paper also investigates other topologies of telerobotic systems when the number of robots involved is more than two resulting in a multilateral architecture. The multilateral architectures can allow for either collaboration between several operators to contact a joint task or can allow collaboration between several remote robotic arms to interact when and if the task cannot be conducted using one follower robotic systems. Another application of multilateral architectures is explained to be expert in the loop training of novice clinicians (such as novice surgeons). The paper highlighted that with the use of advanced architectures, the autonomy of task conduction can be shared between the human operator and the machine intelligence.

The concept of autonomy and various levels of autonomy were explained in detail in this paper. Various examples of shared autonomy were introduced, such as when autonomous agents compensate for the natural tremor of the surgeon's hands, or compensate for organ motions, or provide virtual fixtures to generate either a forbidden region or guiding force field to help the surgeon enhance the quality of the outcome. We have also discussed the use of advanced machine learning methods for augmenting the intelligence of surgical robots and sharing various levels of autonomy between the operator and the machine. In the end, we have categorized and discussed more diverse applications of telerobots, which can form the future of telemedicine in modern healthcare.

To summarize, this review paper provided and updated a comprehensive analysis of the literature and discussed the challenges and future directions of research and development of this technology. We believe this review article can raise awareness of various sectors, including stakeholders and policymakers, to exploit the potential of this technology and boost our healthcare system of tomorrow.

**Author Contributions:** Conceptualization, all authors; methodology, M.T.P., S.F.A., and A.L.; validation, M.T.P., S.F.A., and A.L.; investigation, all authors; resources, all authors; data curation, all authors; writing—original draft preparation, all authors; writing—review and editing, all authors; visualization, all authors; supervision, S.F.A., and A.L.; project administration, A.L.; funding acquisition, S.F.A. All authors have read and agreed to the published version of the manuscript.

**Funding:** This work was supported in part by the National Science Foundation (NSF) under awards #2031594 and 2037878.

**Institutional Review Board Statement:** Not applicable.

**Informed Consent Statement:** Not applicable.

**Data Availability Statement:** Data sharing not applicable.

**Conflicts of Interest:** The authors declare no conflict of interest..

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
