# Peer review of "Review of Advanced Medical Telerobots"

_applsci, doi:10.3390/app11010209_

Round 1

Reviewer 1 Report

This is a good review paper on advanced medical telerobots. Here are some suggestions to improve the quality of the paper.

  1. In most sections, everything is explained with text. There are a few figures showing a general configuration of some systems. It is better to include some block diagrams to show how the control system works. This is very important in the Bilateral Teleoperation section where different control systems are explained. Reading text about a control system is time consuming and at the end, does not give you a clear understanding of how the system works.
  2. Some aspects of the teleoperation systems are explained, however, how these features are related to the stability and transparency of the system are not clearly explained. For example, scaling the motion up or down is explained, scaling the force up or down is explained, but how combination of these two might make the system unstable is not discussed. Motion and force are coupled in a dynamic system. You cannot deal with them separately. There should be a discussion on how these different features might affect the other ones. For example, can we scale up both motion and force? What are the advanced control systems designed to cope wth these problems?
  3. The concept of virtual fixtures is explained, but again, how the stability of the system can be maintained while using such forbidden region virtual fixture is not discussed. Can we define any type of virtual fixtures for the system without sacrificing the stability? Can virtual wall be highly rigid? Should the virtual wall be friction-less or having high friction? How these factors change the stability or performance of the system?
  4. What is the vertical axis in figure 4? Figure 4 is a nice figure, but, when you  are not going to show values or magnitudes on the vertical or horizontal axis, they do not need to be present. The figure is colour coded and do not need axes.
  5. The paper discusses the robotic systems that work in MRI environment, but the most successful MR-compatible Teleoperated robotic system for neurosurgery, neuroArm (which is in clinical use) is not mentioned. The neuroArm system provides haptic feedback.     

Author Response

The authors would like to sincerely thank the reviewers for their insightful comments and suggestions. In response, we have included new details in this revision, that we highlighted in red in the manuscript.

This is a good review paper on advanced medical telerobots. Here are some suggestions to improve the quality of the paper.

In most sections, everything is explained with text. There are a few figures showing a general configuration of some systems. It is better to include some block diagrams to show how the control system works. This is very important in the Bilateral Teleoperation section where different control systems are explained. Reading text about a control system is time consuming and at the end, does not give you a clear understanding of how the system works.

We agree with and thank you for this remark. We have added two figures in this section:

  • one recalling the standard 2,3,4 channel bilateral teleoperation loop (see Fig 1)
  • one showing the main principle of Wave Variable Transformation (see Fig 2)
  • one showing the main principle of Time-Domain Passivity Control (see Fig 3)

Some aspects of the teleoperation systems are explained, however, how these features are related to the stability and transparency of the system are not clearly explained. For example, scaling the motion up or down is explained, scaling the force up or down is explained, but how combination of these two might make the system unstable is not discussed. Motion and force are coupled in a dynamic system. You cannot deal with them separately. There should be a discussion on how these different features might affect the other ones. For example, can we scale up both motion and force? What are the advanced control systems designed to cope wth these problems?

The balance issue between transparency and stability has always been at the heart of Telerobotics. We initially did not focus on it to avoid losing readers in mathematical complex argument, which would have extended much the lenght of this paper. Nevertheless, we agree with the reviewer about the interest of recalling this main issue. We then integrated at the end of the introduction section a new subsection (1.5) (starting at line 186).

The concept of virtual fixtures is explained, but again, how the stability of the system can be maintained while using such forbidden region virtual fixture is not discussed. Can we define any type of virtual fixtures for the system without sacrificing the stability? Can virtual wall be highly rigid? Should the virtual wall be friction-less or having high friction? How these factors change the stability or performance of the system?

We also thank you for this remark. Virtual fixtures were introduced only for their pros, not their cons. To sensitize the readers to these important aspects, we added a paragraph starting at line 517. We based it on the review provided in 2014 by Bowyer et al, that details these stability issues for each kind of virtual fixture.

What is the vertical axis in figure 4? Figure 4 is a nice figure, but, when you are not going to show values or magnitudes on the vertical or horizontal axis, they do not need to be present. The figure is colour coded and do not need axes.

We agree and removed the axes.

The paper discusses the robotic systems that work in MRI environment, but the most successful MR-compatible Teleoperated robotic system for neurosurgery, neuroArm (which is in clinical use) is not mentioned. The neuroArm system provides haptic feedback.

We apologize for this oversight. We added it in the section 3.1 citing the most famous robots on lines 871 and following.

We hope these modifications will answer your questions and satisfy your requests.

Best regards

Reviewer 2 Report

This is a review (survey) paper about telerobotics and medicine. The authors explain the concept, provide historical context, discuss sensory feedback, topology, medical rehabilitation, autonomy among others. 

I found the paper to be informative and comprehensive. The flow of the paper is very smooth and the paper covers important research that has been done in the fields of telerbotics medical procedures. Moreover it stresses the importance of teleoperations in medicine going forward and in light of the current Covid-19 pandemic. 

Few minor suggestions

  • line 23 reference [4, 5] are medical references (the authors mention them as non-medical) 
  • more references on auditory feedback in surgeries if possible (the authors do mention the resources are scarce) - line 251
  • line 442 - pick most relevant references from ref 207 and list them there rather than mentioning to look up references therein. 
  • need references when talking about Qos for e.g. lines 471 and 472
  • line 640 - haptic-enabled negotiation rather than haptics-enabled. 

Author Response

This is a review (survey) paper about telerobotics and medicine. The authors explain the concept,
provide historical context, discuss sensory feedback, topology, medical rehabilitation, autonomy
among others.
I found the paper to be informative and comprehensive. The flow of the paper is very smooth and
the paper covers important research that has been done in the fields of telerbotics medical
procedures. Moreover it stresses the importance of teleoperations in medicine going forward and
in light of the current Covid-19 pandemic.

The authors would like to sincerely thank the reviewers for their insightful comments and
suggestions. In response, we have included new details in this revision, that we highlighted in red
in the manuscript.

Few minor suggestions
• line 23 reference [4, 5] are medical references (the authors mention them as non-medical)

Thanks for your observation. We replaced them with a review dealing with any kind of applications,
not only medical ones, to avoid adding a new reference in the long list of this paper.

• more references on auditory feedback in surgeries if possible (the authors do mention the
resources are scarce) - line 251

We added two references in this paragraph to illustrate possible applications and also the following
sentence: “This kind of feedback has nevertheless interesting applications in basic surgical training
contexts, as demonstrated in [121]”.

• line 442 - pick most relevant references from ref 207 and list them there rather than
mentioning to look up references therein.

To avoid listing the numerous interesting references in the cited paper, which would artificially grow
our reference list, we preferred removing the « and references therein » terms and let readers read
this cited paper on their own.

• need references when talking about Qos for e.g. lines 471 and 472

We added two papers dealing with this issue in line 604 (see, for instance [223,224]).

• line 640 - haptic-enabled negotiation rather than haptics-enabled.

We corrected this error.

We hope these modifications will answer your questions and satisfy your requests.
Best regards

Reviewer 3 Report

This is a well-written paper with an impressive number of references reviewed.  Nice work. 

There are a few typos and items that need to be improved (like line 592).   

There are a few other areas that could be included/improve upon. These are suggestions to improve the depth of the paper. 

There are several important telesurgical applications and advancement that could be included:  

This reference, which I found very interesting, was on telestening operations.  If the authors find it useful, perhaps it could also be added: 

Madder, R.D.; VanOosterhout, S.; Mulder, A.; Bush, J.; Martin, S.; Rash, A.J.; Tan, J.M.; Parker, J.L.; Kalafut, A.; Li, Y.; others. Network latency and long-distance robotic telestenting: Exploring the potential impact of network delays on telestenting performance. Catheterization and Cardiovascular Interventions. Wiley Online Library, 2019.  There may be even more work from that group.

Regarding Instability Challenge (~line 347).  There are many groups and in fact, entire review articles written on time-delay mitigation.  It would be interesting to see if some basic reviews on this topic can be pointed to.  There are approaches that use machine learning methods, specifically neural networks and even time series prediction. Other researches focus on the control systems aspects of time delay and instability. In fact, some recent work also points to how scaling down movements leads to better performance for time-delayed systems.  A paragraph or so pointing to this work would improve that discussion.

In terms of single-leader multi-follower type systems, there has been recent interest in record and playback of surgical movement for training purposes and also for creating datasets for AI systems to learn from.  This might be of interest to your readers.  See if you can find some work of interest there.

Multi-leader/Multi-follower seems sparse and unclear in terms of applications.  Please clarify this section's introduction as to what the potential applications are.

The section on Autonomous levels is missing a section on camera control.  Currently, in robotic surgery, the surgeon is expected to control both the tools and the camera.  There have been a recent surge in view point automation and camera control algorithms for the automation of view points for the da Vinci system.  It would be nice to include some review papers and some recent papers in this emerging (albeit low-hanging fruit) area.

Overall very nice review and I like the forward-thinking robot teleoperation topologies reviewed. 

Author Response

The authors would like to sincerely thank the reviewers for their insightful comments and suggestions. In response, we have included new details in this revision, that we highlighted in red in the manuscript.

This is a well-written paper with an impressive number of references reviewed.  Nice work.

here are a few typos and items that need to be improved (like line 592).

We thank you for these encouraging comments. We are sorry for this typo that escaped our attention. We fixed it.

There are a few other areas that could be included/improved upon. These are suggestions to improve the depth of the paper. 

There are several important telesurgical applications and advancement that could be included:

This reference, which I found very interesting, was on telestening operations. If the authors find it useful, perhaps it could also be added: 

Madder, R.D.; VanOosterhout, S.; Mulder, A.; Bush, J.; Martin, S.; Rash, A.J.; Tan, J.M.; Parker, J.L.; Kalafut, A.; Li, Y.; others. Network latency and long-distance robotic telestenting: Exploring the potential impact of network delays on telestenting performance. Catheterization and Cardiovascular Interventions. Wiley Online Library, 2019.  There may be even more work from that group.

Regarding Instability Challenge (~line 347). There are many groups and in fact, entire review articles written on time-delay mitigation.  It would be interesting to see if some basic reviews on this topic can be pointed to. There are approaches that use machine learning methods, specifically neural networks and even time series prediction. Other researches focus on the control systems aspects of time delay and instability. In fact, some recent work also points to how scaling down movements leads to better performance for time-delayed systems.  A paragraph or so pointing to this work would improve that discussion.

The balance issue between transparency and stability has always been at the heart of Telerobotics. We initially did not focus on it to avoid losing readers in mathematical complex arguments, which would have extended much the length of this paper. Nevertheless, we agree about the interest of recalling this main issue. We then integrated at the end of the introduction a new subsection (1.5) (starting at line 204).

In terms of single-leader multi-follower type systems, there has been recent interest in record and playback of surgical movement for training purposes and also for creating datasets for AI systems to learn from.  This might be of interest to your readers.  See if you can find some work of interest there.

We agree with the interest of this opportunity. We added two paragraphs accordingly (starting at line 186).

Multi-leader/Multi-follower seems sparse and unclear in terms of applications.  Please clarify this section's introduction as to what the potential applications are.

To illustrate this kind of telerobot, we have added a mention about    the da VInci system, and  about the potential other applications including  cooperation between several surgeons with more than two tools, with operators potentially being distant from each other and from the patient. (see section 2.3.3, lines 725-733).

The section on Autonomous levels is missing a section on camera control.  Currently, in robotic surgery, the surgeon is expected to control both the tools and the camera.  There have been a recent surge in view point automation and camera control algorithms for the automation of view points for the da Vinci system.  It would be nice to include some review papers and some recent papers in this emerging (albeit low-hanging fruit) area.

We have incorporated this important feature in the paper in lines 982-1003.

We hope these modifications will answer your questions and satisfy your requests.

Best regards